# Mutations accumulated in the Spike of SARS-CoV-2 Omicron allow for more efficient counteraction of the restriction factor BST2/Tetherin

**Yuhang Shi, Sydney Simpson[¤a], Yuexuan Chen, Haley Aull, Jared Benjamin[¤b], Ruth Serra-Moreno** [*]

Microbiology and Immunology, University of Rochester Medical Center, Rochester, New York, United States of America

¤a Current address: QuidelOrtho, Rochester, New York, United States of America
¤b Current address: Icahn School of Medicine, Mount Sinai, New York, New York, United States of America
* ruth_serra-moreno@urmc.rochester.edu

**Data Availability Statement:** All relevant data are in the manuscript and its supporting information files.

## Abstract

BST2/Tetherin is a restriction factor with broad antiviral activity against enveloped viruses, including coronaviruses. Specifically, BST2 traps nascent particles to membrane compartments, preventing their release and spread. In turn, viruses have evolved multiple mechanisms to counteract BST2. Here, we examined the interactions between BST2 and SARS-CoV-2. Our study shows that BST2 reduces SARS-CoV-2 virion release. However, the virus uses the Spike (S) protein to downregulate BST2. This requires a physical interaction between S and BST2, which routes BST2 for lysosomal degradation in a Clathrin- and ubiquitination-dependent manner. By surveying different SARS-CoV-2 variants of concern (Alpha-Omicron), we found that Omicron is more efficient at counteracting BST2, and that mutations in S account for its enhanced anti-BST2 activity. Mapping analyses revealed that several surfaces in the extracellular region of BST2 are required for an interaction with the Spike, and that the Omicron variant has changed its patterns of association with BST2 to improve its counteraction. Therefore, our study suggests that, besides enhancing receptor binding and evasion of neutralizing antibodies, mutations accumulated in the Spike afford more efficient counteraction of BST2, which highlights that BST2 antagonism is important for SARS-CoV-2 infectivity and spread.

## Author summary

BST2/Tetherin is a potent antiviral factor that prevents the egress of multiple enveloped viruses. In turn, viruses have evolved mechanisms to circumvent this block. Here, we found that SARS-CoV-2 primarily uses the Spike protein to promote the lysosomal degradation of BST2, thus, removing it from sites of virion assembly and facilitating virus release. When analyzing several SARS-CoV-2 variants of concern, we found that Omicron is more efficient at counteracting BST2, to the point where its replication is barely

**Funding:** These studies have been funded through a University of Rochester Research Award to RSM (OP212320) and an Empire State Development NYFirst to RSM (# 134,351). The funders had no role in study design, data collection and analysis, decision to publish, or preparation of the manuscript.

**Competing interests:** The authors have declared that no competing interests exist.

impacted by this restriction factor. Subsequent studies identified mutations accumulated in the Omicron Spike as responsible for improving an interaction between Spike and BST2. However, the surfaces in BST2 required for this association differ between Wuhan and Omicron. Remarkably, increased Spike-BST2 binding is associated with enhanced BST2 downregulation. Therefore, these observations suggest that, in addition to enhancing receptor binding and immune evasion, mutations in the Spike afford more efficient counteraction of BST2, highlighting that BST2 antagonism is important for SARS-CoV-2 infectivity.

## Introduction

Bone marrow stromal antigen 2 (BST2, also known as Tetherin and CD317) is a type I interferon inducible cellular factor that restricts the release of budding enveloped virus particles from infected cells [1–5]. BST2 is a type II transmembrane protein that contains an N-terminal cytoplasmic tail followed by a transmembrane domain (TM), a coiled-coil ectodomain, and a C-terminal glycosylphosphatidylinositol (GPI) anchor that attaches the C-terminal domain to cholesterol-enriched locations on the outer leaflet of cellular membranes [6]. BST2 forms oligomers and localizes at the plasma membrane, endosomal compartments, endoplasmic reticulum, and the *trans*-Golgi network (TGN) [6,7]. Both the TM and the GPI anchor form crosslinks between cellular membranes and budding viral particles, preventing virion egress and the subsequent spread of infection [8,9]. Aside from accruing virions at the cell surface, particles trapped by BST2 are routed to endosomal compartments for lysosomal clearance [10].

BST2 was initially identified as a membrane protein downregulated by Kaposi Sarcoma Herpes Virus (KSHV) and Human Immunodeficiency Virus (HIV), suggesting a potential antiviral function [11]. These findings led to the later discovery that BST2 is indeed a restriction factor against HIV that tethers HIV virions to the host cell membrane [2,4]. Since that discovery, BST2 has been shown to inhibit the spread of multiple enveloped viruses including other retroviruses, arenaviruses, influenza viruses, herpesviruses, filoviruses, and several coronaviruses [3,4,9,12–17]. For instance, BST2 was found to decrease SARS-CoV-1 infectivity [16,17], sequester human coronavirus 229E particles at the plasma membrane and in intracellular compartments [15], and restrict Porcine Endemic Diarrhea coronaVirus (PDEV) by promoting the ubiquitination and degradation of the virus nucleocapsid [18]. To counteract BST2, viruses have evolved mechanisms to either downregulate or disable this host restriction factor. The most notable example of this is that of the primate lentiviruses. Whereas the majority of Simian Immunodeficiency Viruses (SIVs) use the virus protein Nef to route BST2 for lysosomal degradation [19,20], HIV-1 and HIV-2 evolved alternative strategies, since human BST2 harbors a deletion in the region targeted by Nef. Specifically, HIV-1 uses Vpu to sequester and/or promote the degradation of BST2 by both proteosomes and lysosomes [21,22], while HIV-2 evolved the envelope glycoprotein (Env) to remove BST2 from sites of virion assembly [5]. Similar to HIV-2 Env, SARS-CoV-1 was found to use its Spike glycoprotein (S) to counteract BST2. However, the mechanism of counteraction is different. While HIV-2 Env sequesters BST2 in the *trans*-Golgi network [5], the Spike promotes the lysosomal clearance of BST2 [17]. In addition to S, ORF7a was also reported to inhibit BST2 antiviral activity by interfering with BST2 glycosylation and subsequent function [16]. Recent studies also identified ORF7a and S as putative BST2 antagonists in SARS-CoV-2, although their underlying mechanism of action remains to be elucidated [23–27]. Here, we investigated the role of BST2 as a

restriction factor against SARS-CoV-2 infection and if SARS-CoV-2, in turn, counteracts BST2.

SARS-CoV-2 emerged late in 2019 and is the causative agent of COVID-19. SARS-CoV-2 is a β-coronavirus and the seventh coronavirus known to infect humans [28,29]. While most coronaviruses cause mild upper respiratory tract infections, two other coronaviruses, namely SARS-CoV-1 and MERS-CoV, can cause severe respiratory distress and even death. Coronaviruses encode for many accessory and non-structural proteins, while only a small portion of their genome is devoted to the structural proteins: Spike (S), Envelope (E), Membrane (M), Nucleocapsid (N), and, in some coronavirus species, Hemagglutinin Esterase (HE). SARS-CoV-2 infects cells in the respiratory and gastrointestinal tract using the S protein as anti-receptor and the angiotensin converting enzyme 2 (ACE2) as a receptor, which is the main determinant for cell tropism [28,30,31]. After this initial binding, the TM protease serine 2 (TMPRSS2), a protease located at the host cell membrane, activates the S protein by cleaving it into two subunits, S1 and S2 [32–34]. The S2 subunit then mediates fusion between the virus envelope and the host cell membrane [34,35]. After virus entry, SARS-CoV-2 remodels cellular membranes to generate replication organelles for genome replication, transcription, and virus protein synthesis. Once the structural proteins are synthesized, they are translocated to the ER membrane and subsequently transferred through the secretory pathway to the ER-Golgi Intermediate Compartment (ERGIC), where virion assembly takes place. From the ERGIC, virions are trafficked through the *trans*-Golgi network to the plasma membrane and are released via exocytosis [36]. Notably, BST2 can be found in several compartments including ER, Golgi and the ERGIC, which could have an impact on SARS-CoV-2 budding. Here, we confirm that BST2 restricts SARS-CoV-2 infectious particle production. However, similar to SARS-CoV-1, the virus uses S to antagonize BST2. Specifically, S interacts with BST2 to route this restriction factor to lysosomes for degradation in a Clathrin- and ubiquitination-dependent manner. Hence, the virus removes BST2 from sites of assembly, restoring viral release. The SARS-CoV-2 Spike contains regions of variation that serve as hot-spots for the accumulation of mutations that account, in part, for the differences in infectivity, transmission and pathogenicity observed among variants of concern [37–40]. Hence, we studied whether mutations in S affect BST2 antagonism. Remarkably, the Spike of emerging SARS-CoV-2 variants, particularly Omicron, more efficiently downregulate BST2. Therefore, these findings suggest that in addition to enhanced attachment and evasion of neutralizing antibodies, mutations in S facilitate escape from BST2 restriction, highlighting that BST2 antagonism is important for SARS-CoV-2 infectivity and spread.

## Results

### *BST2* is induced in response to SARS-CoV-2 and in turn SARS-CoV-2 downregulates BST2 protein levels

To determine whether BST2 is relevant to SARS-CoV-2 biology, we assessed if it is induced in response to SARS-CoV-2 infection. For this, ACE2-expressing A549 human lung epithelial cells were infected with the Hong Kong (HK) variant of SARS-CoV-2 or lentiviral-like particles (mock) at a multiplicity of infection (MOI) of 1 and *BST2* expression was determined by RT-qPCR 6 hours later. As expected, a significant upregulation of *BST2* was observed in the presence of SARS-CoV-2 (**Fig 1A**), reflecting that, as part of their innate antiviral arsenal, cells respond to this virus by inducing *BST2*. Next, we investigated the antiviral potential of BST2. For this, we chose HEK293T cells because they do not express detectable levels of BST2, which allows to examine BST2 antiviral activities by comparing parental with BST2-engineered cells, making this system a gold standard for mechanistic studies on this antiviral factor [2–

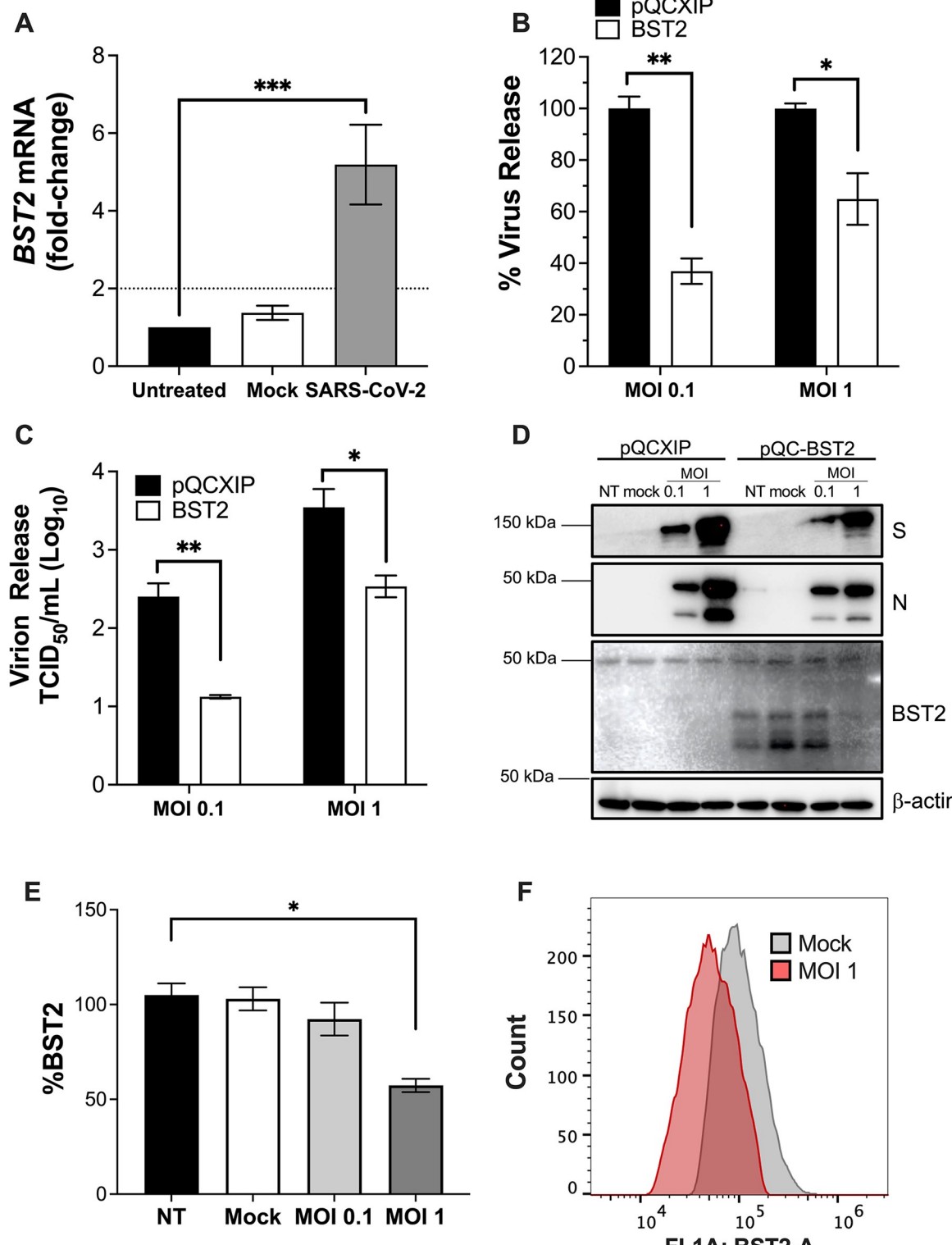

**Fig 1. BST2 inhibits SARS-CoV-2 virion release but the virus downregulates BST2.** (**A**) A549-ACE2 cells were not infected (untreated), infected with lentiviral-like particles (mock), or infected with SARS-CoV-2 HK at MOI = 1. Six hours later, the mRNA levels of *BST2* were measured by RT-qPCR. (**B** & **C**) HEK293T-ACE2 cells stably expressing pQCXIP or pQCXIP-BST2 were infected with SARS-CoV-2 HK at MOI = 0.1 or 1. Nontreated (NT) and mock-infected cells were included as controls. Twenty-four hours post-infection, the supernatants were collected to examine virion release by N-ELISA, which was expressed as the percentage of virus release

(B), and by $TCID_{50}$ (C). (**D**) Cells were also harvested to measure the levels of BST2 and virus proteins by western blot. (**E**) Relative BST2 expression was calculated by densitometry analyses, normalized to actin, and expressed as the percentage of BST2. (**F**) The levels of BST2 were also measured by flow cytometry in mock-infected cells and cells infected with SARS-CoV-2 HK at MOI = 1. *: $p < 0.05$, **: $p < 0.01$, ***: $p < 0.001$. Blots are representative of 3 biological replicates. Data correspond to the mean and SEM of 3 independent experiments.

4,19,20,41–44]. HEK293T-ACE2$^+$ cells stably expressing an empty retroviral vector (pQCXIP) or BST2 (pQCXIP-BST2) were infected with SARS-CoV-2 HK at MOI = 0.1 or 1. Twenty-four hours post-infection, the culture supernatants were recovered to assess virion production in the presence and absence of BST2, which was measured by SARS-CoV-2 N-specific ELISA and infectivity assays ($TCID_{50}$). Additionally, the infected cells were harvested for western blot analyses. In line with findings for other enveloped viruses, BST2 significantly reduced the number of virions released to the culture supernatant (**Fig 1B**). Consistent with this decrease in particle release, the infectivity of the culture supernatants derived from cells expressing BST2 significantly declined (**Fig 1C**). However, the impact of BST2 on virion production and infectivity was reduced at the high MOI (**Fig 1B** and **1C**), suggesting that the virus can circumvent this block. This was confirmed when analyzing infected cells by western blot, showing that at MOI = 1 the virus downregulates BST2 (**Fig 1D**). Densitometry quantifications of BST2 from three independent experiments revealed that SARS-CoV-2 HK causes a ~40% reduction in the BST2 protein levels (**Fig 1E**), and this was verified by flow cytometry (**Fig 1F**). Overall, these results indicate that BST2 restricts SARS-CoV-2 virion release, but the virus has evolved to circumvent this block by reducing BST2.

## SARS-CoV-2 Omicron has enhanced anti-BST2 activity

Since BST2 affects a broad spectrum of enveloped viruses, it is often targeted by virus pathogens to enhance virion production, and this has resulted in an arms-race between viruses and BST2 [45,46]. In fact, BST2 is under positive selection [47]. This fact prompted us to investigate whether recent SARS-CoV-2 variants exhibit differences in their abilities to antagonize BST2 compared to earlier isolates. To test this, we performed parallel infections with SARS-CoV-2 HK and Omicron. Here, we chose A549-ACE2$^+$ cells stably expressing pQCXIP or BST2, because they are more physiologically relevant to SARS-CoV-2 infection than HEK293T cells. Cells were infected at MOI = 0.1 or 1. Twenty-four hours later, cell lysates and supernatants were recovered to assess BST2 levels and infectious virion production, as detailed above. Similar to our findings in HEK293T cells, HK downregulated BST2, but in this system, this downregulation was noticeable even when cells were infected at MOI = 0.1 (**Fig 2A**; see pQC-BST2 lanes). Remarkably, we found that SARS-CoV-2 Omicron is better equipped at downregulating BST2 and, consequently, particle release is negligibly impacted by this restriction factor (**Fig 2A** and **2B**). Of note, unlike parental HEK293T cells, where BST2 is undetectable, we observed endogenous BST2 expression in parental A549-ACE2$^+$ cells (**Fig 2A**; see pQCXIP lanes). Although these levels are lower than in the cells engineered to stably express BST2, downregulation of BST2 is still noticeable for both HK and Omicron, with Omicron causing more downregulation of BST2 (**Fig 2A**; see pQCXIP lanes). These findings led us to investigate the degree of BST2 expression in the stable cells compared to BST2 induction afforded by interferon (IFN) stimulation, so we could ascertain the physiological relevance of our BST2 engineered cells to SARS-CoV-2. Our assays revealed that BST2 levels are comparable between A549-ACE2$^+$ cells treated with IFNα and the stably expressing BST2 cells (**S1 Fig**).

In view of these results, we next assessed the ability of SARS-CoV-2 HK and Omicron to counteract BST2 when its expression is induced by IFNα. For this, A549-ACE2$^+$ cells were infected with SARS-CoV-2 HK or Omicron at MOIs = 0.1, 1 or 5. SARS-CoV-2 viral like

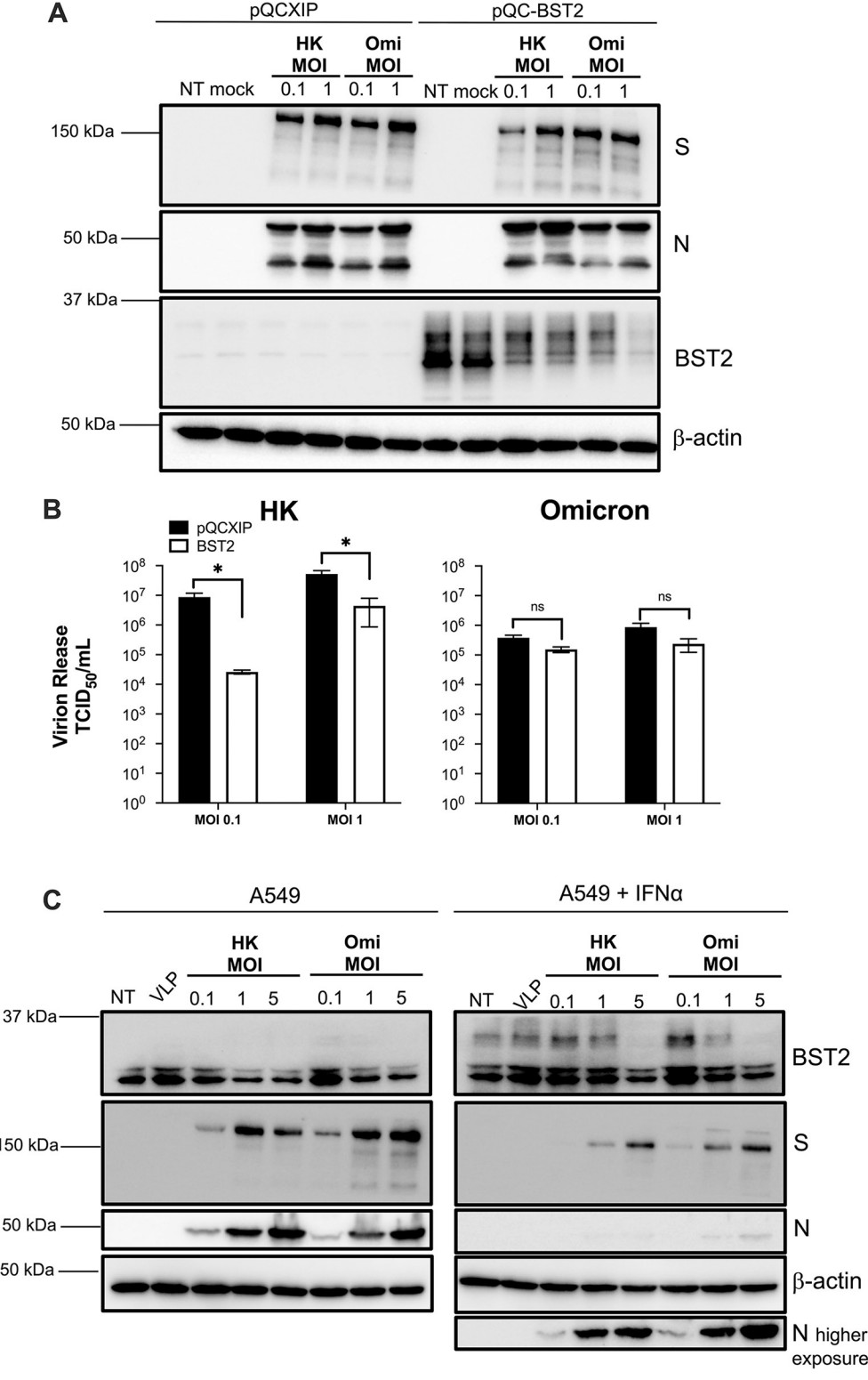

**Fig 2. SARS-CoV-2 Omicron has improved anti-BST2 activity.** (**A** & **B**) A549-ACE2 cells stably expressing pQCXIP or pQCXIP-BST2 were infected with SARS-CoV-2 HK or Omicron variants at MOI = 0.1 or 1. Twenty-four hours post-infection, the levels of BST2 and virus proteins were examined by western blot (A), and virion production was measured by TCID$_{50}$ (B). (**C**) A549-ACE2 cells were infected with SARS-CoV-2 HK or Omicron at MOI = 0.1, 1 or 5. Uninfected cells (NT) and cells infected with SARS-CoV-2 VLPs at MOI ~1 were included as controls. One-hour post-

infection, cells were treated with DMSO or IFNα. Twenty-four hours later, the levels of BST2 and virus proteins were examined by western blot. *: $p < 0.05$, ns: not significant. Blots are representative of 3 biological replicates. Data correspond to the mean and SEM of 3 independent experiments.

particles (VLPs) were used to examine for effects on BST2 caused by productive and abortive infections. One hour later, cells were washed and treated with either DMSO or IFNα, as previously described [15,48]. Twenty-four hours post-infection, cells were harvested and analyzed by western blot. Consistent with our experiments in A549-ACE2-BST2 cells, SARS-CoV-2 productive infection downregulated IFN-induced BST2 (Fig 2C). In line with our previous findings, Omicron caused a more dramatic downregulation of IFN-induced BST2 than HK. While BST2 downregulation only became evident in cells infected with HK at MOI = 5, Omicron caused a noticeable downregulation of the protein at MOI = 1 (Fig 2C; right panel). Similar results were observed in parental A549-ACE2$^+$ cells treated with DMSO, although under these conditions BST2 downregulation is observed at lower MOIs (Fig 2C; left panel). Remarkably, we noticed that BST2 induction was augmented in infections with Omicron at MOI = 0.1, which is consistent with recent findings showing that Omicron enhances interferon promoter activity [49–51]. So, while the overall BST2 levels in parental cells infected with HK and Omicron at MOI = 1 appear similar (Fig 2C; left panel compare lanes 4 and 7), the downregulation afforded by Omicron is greater, since the initial, endogenous levels of BST2 are more elevated for Omicron infections (Fig 2C; left panel, compare lanes 3 and 6). In summary, these results demonstrate that SARS-CoV-2-mediated downregulation of BST2 is relevant to infection, and that Omicron is more efficient at counteracting BST2 than early isolates.

To assess if enhanced BST2 antagonism is a phenotype exclusive of Omicron, additional assays were performed with other variants of concern (Alpha, Beta, and Delta). Our data show that more recent isolates exhibit enhanced BST2 downregulation and a parallel increase in particle release. These observations were recapitulated in both HEK293T-ACE2$^+$ and A549-ACE2$^+$ cells stably expressing BST2 (S2 Fig). Hence, these findings indicate that in addition to enhancing attachment and evading host immune responses, mutations accumulated in emerging SARS-CoV-2 strains facilitate escape from BST2 restriction.

## SARS-CoV-2 uses the Spike protein to downregulate BST2

To elucidate how SARS-CoV-2 downregulates BST2, we used an unbiased approach in which we examined how each SARS-CoV-2 protein affects BST2. For this, we utilized an expression library generated by the Krogan laboratory [52]. HEK293T-ACE2 cells stably expressing BST2 were transiently transfected with constructs coding for each SARS-CoV-2 protein. HIV-1 Vpu and SARS-CoV-1 S (SARS-1 S) were used as positive controls, since they both promote the degradation of BST2 [2,4,17], and GST-HA was included as a negative control (Fig 3A). Similar to SARS-CoV-1, the Spike of SARS-CoV-2 caused a significant reduction in BST2 levels (Fig 3A). Additionally, ORF7a altered the SDS-PAGE migration pattern of BST2, by causing the accumulation of a lower molecular band of the protein, which likely represents BST2 with limited post-translational modifications (Fig 3A; right panel). Besides the Spike, previous work with SARS-CoV-1 identified ORF7a as another BST2 antagonist. While S promoted the downregulation of BST2 via lysosomal degradation [17], ORF7a affected BST2 glycosylation in a manner that low molecular BST2 isoforms–which have lower antiviral activity–accumulated in the cell [16,17]. Hence, the phenotype we are detecting here is in line with those observations. Our findings are also consistent with recent work that identified ORF7a and S of SARS-CoV-2 as putative BST2 antagonists [23–27]. Whereas both proteins were found in complexes

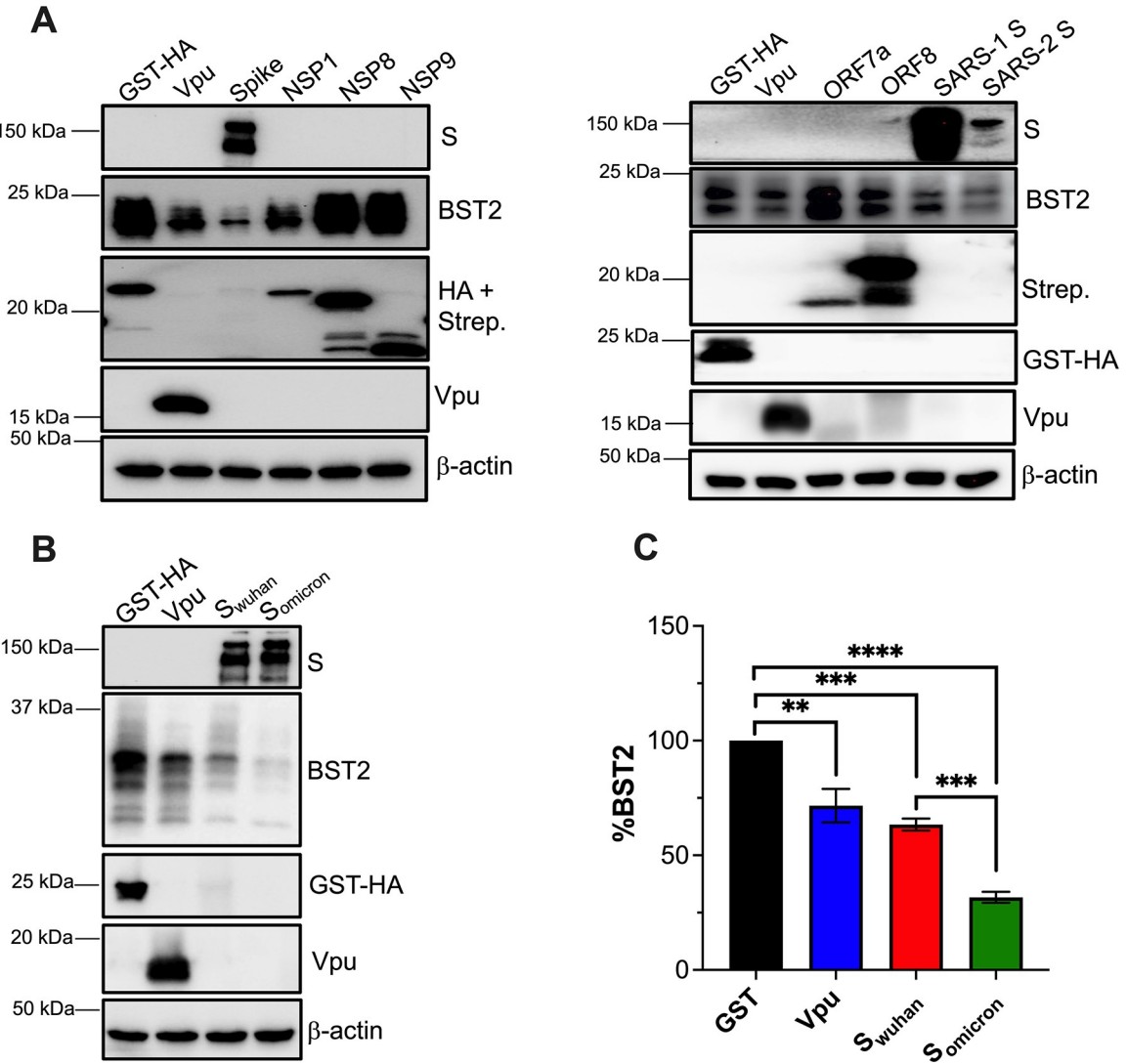

**Fig 3. The Spike protein is the main BST2 antagonist in SARS-CoV-2.** (**A**) HEK293T-ACE2 cells stably expressing BST2 were transfected with plasmids coding different proteins of SARS-CoV-2 Wuhan. As a negative control, a construct encoding GST was included. As positive controls, constructs coding for HIV-1 *vpu* or SARS-CoV-1 *S* (SARS-1 S) were included. The levels of BST2, GST, virus proteins and β-actin were measured by western blot. (**B**) HEK293T-ACE2 cells stably expressing BST2 were transfected with plasmids encoding the *Spike* gene of SARS-CoV-2 Wuhan or Omicron. GST and HIV-1 Vpu were included as controls. BST2, GST, Vpu and Spike levels were measured by western blot. (**C**) Relative BST2 expression was calculated by densitometry analyses, normalized to actin, and expressed as the percentage of BST2. **: $p<0.01$, ***: $p<0.001$, ****: $p<0.0001$. Blots are representative of 3 biological replicates. Data correspond to the mean and SEM of 3 independent experiments.

with BST2, only S downregulated this restriction factor–although the underlying mechanism remains unknown [24,26,53]. In addition to the Spike, we found that NSP1 also downregulated BST2, but to a lesser extent than the Spike (**Fig 3A**; left panel). NSP1 is a shutoff protein that favors virus gene expression in detriment of cellular genes by preventing translation of cellular mRNAs and accelerating mRNA decay [54–56]. Therefore, BST2 downregulation by SARS-CoV-2 is afforded by the virus Spike and NSP1 shutoff effects.

Our assays with SARS-CoV-2 variants of concern show that recently emerged strains perform better at counteracting BST2 (**Figs 2 and S2**). The SARS-CoV-2 Alpha-Omicron variants are characterized by the accumulation of mutations in the *Spike* reading frame, which afford

increased affinity for the ACE2 receptor and help the virus escape from neutralizing antibodies [57–60] (see **S3 Fig** for a schematic of the mutations in SARS-CoV-2 S). However, additional substitutions were found in other regions of the genome. Because mutations in S have been studied in more detail, we did not want to overlook other substitutions that might be responsible for improved BST2 antagonism, such as changes in ORF7a or NSP1. Except the Delta variant–which harbors two amino acid substitutions at residues 82 and 121 –Alpha, Beta and Omicron have no amino acid changes in ORF7 (**S4A Fig**). Similarly, NSP1 is well-conserved across the SARS-CoV-2 variants. Only Omicron had an amino acid substitution at position 135, which was reverted to the original residue in subsequent Omicron subvariants (**S4B Fig**). Due to the lack of variation in ORF7a and NSP1 across the variants of concern, we reasoned that SARS-CoV-2 primarily uses the Spike to overcome restriction by BST2, and that mutations accumulated in S account for the improved BST2 antagonism in these strains. We tested this hypothesis by comparing the Spikes of Wuhan and Omicron for their ability to downregulate BST2 from HEK293T cells stably expressing this restriction factor. GST-HA and HIV-1 Vpu were used as controls. Forty-eight hours post-transfection, cells were harvested and the levels of GST-HA, Vpu, Spike and BST2 were examined by western blot. Remarkably, the Spike of Omicron ($S_{omicron}$) acquired improved anti-BST2 activity (**Fig 3B** and **3C**). Similar experiments were performed with the Spikes of other variants of concern, revealing that mutations in the Spikes afford enhanced downregulation of BST2, especially in later variants. For instance, the Alpha Spike achieves 40% downregulation of BST2, while the Omicron Spike can reduce up to 70% of BST2 levels (**S5 Fig**). Hence, these findings confirm that the Spike is the main BST2 antagonist of SARS-CoV-2.

## The Spike interacts with BST2, and this association is more efficient in SARS-CoV-2 Omicron

Studies on SARS-CoV-1 and other enveloped viruses have found that many of these viruses use their envelope glycoproteins to downregulate BST2, and this requires a physical interaction between the viral glycoproteins and BST2 [5,17,41,42]. To investigate if SARS-CoV-2 uses a similar mechanism to overcome restriction by BST2, co-immunoprecipitation (co-IP) assays were performed in HEK293T cells transiently transfected with BST2. SARS-CoV-1 S and SARS-CoV-2 ORF7a were used as positive controls. Consistent with findings with SARS-CoV-1, an interaction between BST2 and SARS-CoV-2 S was detected (**Fig 4A**; left panel). In agreement with recent findings [26,27], BST2 and SARS-CoV-2 ORF7a were also found to interact (**Fig 4A**; right panel). However, only the Spikes of SARS-CoV-1 and SARS-CoV-2 caused the downregulation of BST2 (**Fig 4A**; see whole cell lysate), reinforcing the notion that the Spike is the main BST2 antagonist in SARS-CoV-2.

To understand how the Omicron Spike affords improved downregulation of BST2, we compared the BST2 binding efficiency between the Wuhan and Omicron Spikes by co-immunoprecipitation and found a significant increase in the levels of Omicron Spike present in the BST2 pulldown fraction (**Fig 4B**; see quantification). This increase in Spike-BST2 interaction is associated with augmented depletion of BST2 (**Fig 4B**; see whole cell lysates). Therefore, mutations in the Spike allow for increased binding to BST2 –or an intermediate factor–which likely enhances BST2 downregulation.

## Multiple surfaces of BST2 are required for Spike-BST2 interaction

To map the domains in BST2 required for the Spike-BST2 association, we generated a series of truncation and point mutants in BST2. Specifically, we generated: (i) a mutant in which the BST2 cytoplasmic tail was truncated (ΔCT), (ii) a mutant in which the transmembrane domain

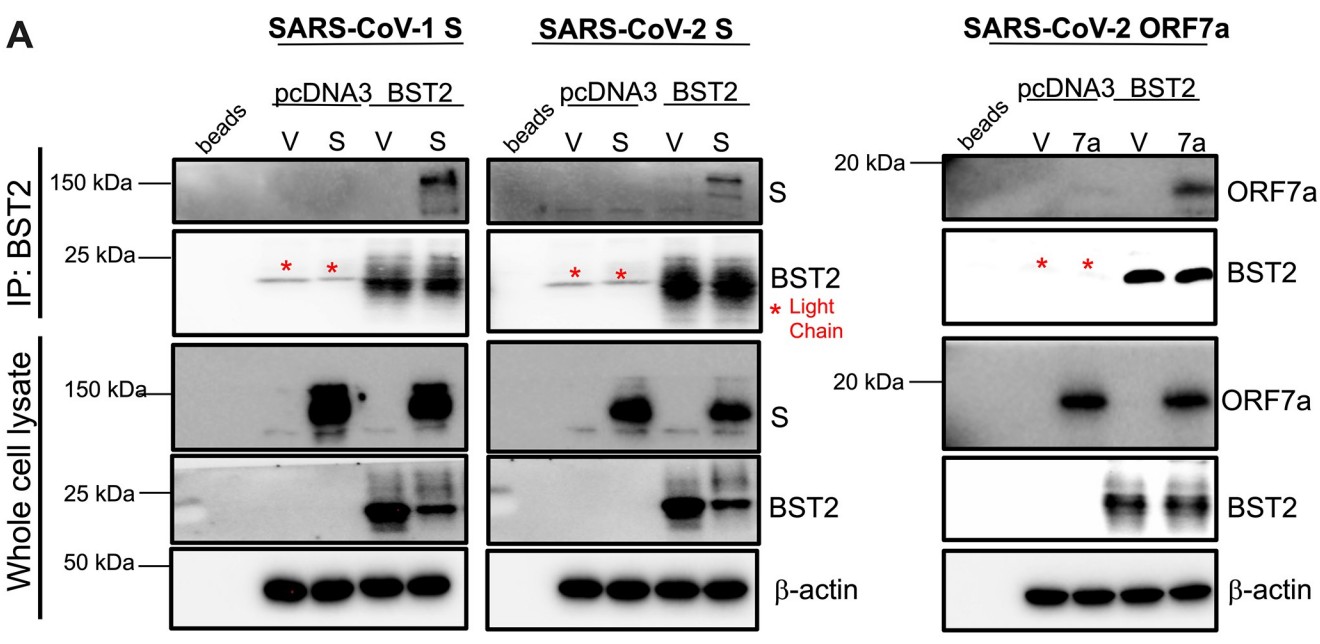

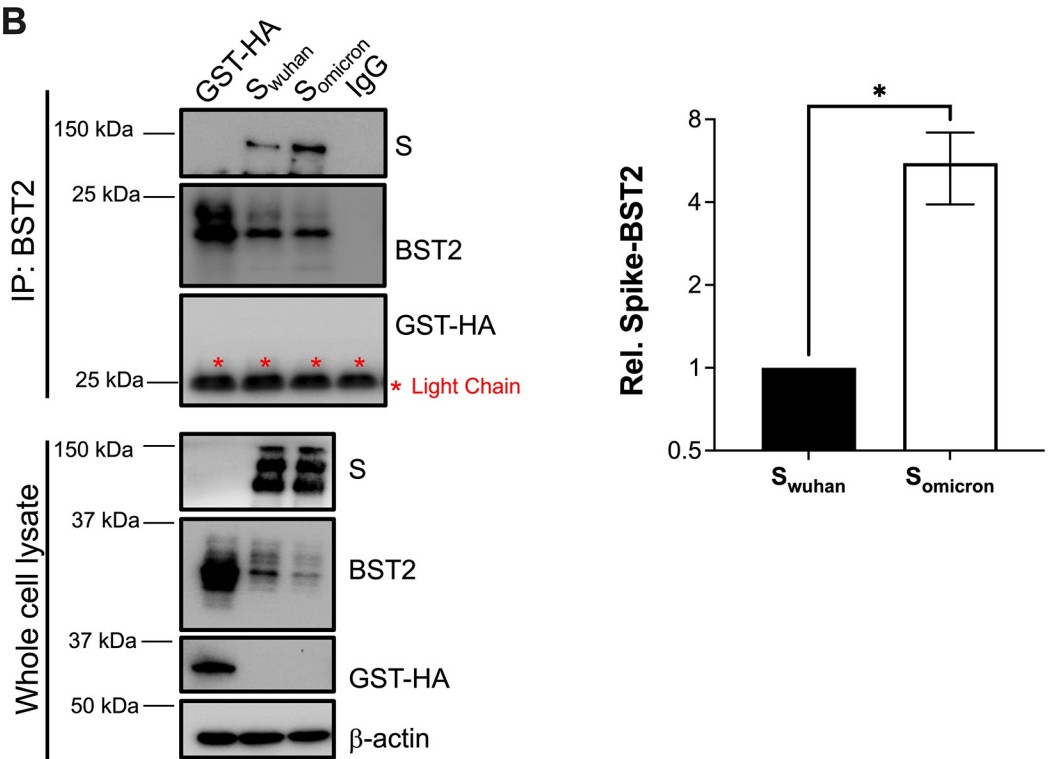

**Fig 4. The Spike of SARS-CoV-2 physically interacts with BST2, and S$_{omicron}$ binds more efficiently to the protein.** (**A**) The interaction between BST2-SARS-CoV-2-S and BST2-SARS-CoV-2-ORF7a were evaluated by co-immunoprecipitation (co-IP). Empty vector (V), SARS-CoV-1 S, and lysis buffer incubated with beads were included as controls. (**B**) The interactions of BST2-S$_{wuhan}$ and BST2-S$_{omicron}$ were examined by co-IP. A plasmid encoding *GST* and beads incubated with antibody (IgG) were included as controls. Graph: relative binding between BST2 and the Spikes was calculated by densitometry analyses from 4 biological replicates. Red asterisks indicate bands corresponding to the light chain of the antibody used in the IP. *: $p < 0.05$. Blots are representative of a minimum of 3 independent experiments. Data correspond to the mean and SEM of 4 independent experiments.

of BST2 was replaced by the transmembrane domain of the human transferrin receptor (TfR). This construct has already been reported to display similar subcellular distribution to wild type BST2 and retains its antiviral properties [9]. (iii) A mutant in which 50 residues of the extracellular domain region–prior to the coiled coil domain–were deleted (ΔEC1), (iv) a mutant with the 54-amino acids of the coiled-coil domain truncated (ΔCC), and (v) a mutant harboring six alanine substitutions in the short region between the coiled-coil domain and the GPI anchor (EC2$_{Ala}$) (see **Fig 5A** for a schematic of BST2 architecture and the constructs generated). These mutants were tested for their ability to interact with the Wuhan and Omicron Spikes by co-IP after co-transfection in HEK293T cells. We need to stress that studying protein-protein interactions of membrane proteins by co-IP can be complicated as improper lysis of cellular membranes can lead to false positives. To address this, we included controls allowing us to detect loss of binding between these membrane proteins, which are detailed below.

All mutants, including wild type BST2, harbored an HA tag in their N-terminus to facilitate immunoprecipitation and to assess overall protein expression. Furthermore, transfections with HIV-1 Vpu were included, since Vpu interacts with BST2 and needs the transmembrane domains of both proteins for binding. Hence, a loss of interaction between Vpu and the TfR-TM mutant is expected [9,19,47]. GFP was included as a negative control because it does not interact with BST2. Here, we did not use the GST-HA control since the BST2 constructs were HA-tagged. It is important to note that the plasmid encoding *vpu* also harbors *GFP* from an Internal Ribosomal Entry Site (IRES), so samples expressing Vpu are also positive for GFP. Forty-eight hours post-transfection, cells were harvested, and lysates were immunoprecipitated using an HA antibody. Membranes were probed for HA, SARS-CoV-2 S, Vpu, and GFP. Internal immunoprecipitation controls were included, such as lysates of transfections with the Spikes and the full-length BST2 incubated with beads but no antibody (beads control), as well as lysis buffer incubated with beads and the HA antibody (IgG control). These controls allowed us to discriminate bands that correspond to any unspecific binding of the Spikes with the beads, the anti-HA antibody heavy and light chains, and/or debris material from the beads, respectively.

Consistent with previous publications [9,19,47], we observed an interaction between HIV-1 Vpu and BST2, and this association was lost when the TM domain of BST2 was deleted or replaced by the transmembrane domain of the transferrin receptor. Accordingly, when the Vpu-BST2 interaction disappeared, no downregulation of BST2 was observed (**Figs 5B, 5C**, **S6B and S6C**). However, even though Vpu associates with the ΔCT mutant, no downregulation of ΔCT-BST2 was detected (**S6B Fig**; see whole cell lysate lanes 5 and 6), which is consistent with the fact that Vpu recruits cellular factors to promote the ubiquitination of Ser and Thr residues in the cytoplasmic tail of BST2, which routes it for degradation [61–63]. As expected, no association between GFP and BST2 was observed, only the light chain of the anti-HA antibody was detected in these blots (**Figs 5B, 5C, and S6**; red asterisks). In agreement with our data in Fig 4, we recapitulated an association between the Wuhan and Omicron Spikes with full-length BST2. However, the interaction was compromised with certain BST2 truncation mutants. Specifically, S$_{wuhan}$ was undetectable in IPs with the ΔCC mutant, and a significant loss in binding was also found with the ΔEC1 mutant. However, the alanine substitutions in the EC2 region, deletion of the cytoplasmic tail or changing the TM domain caused no defect in the ability of S$_{wuhan}$ to interact with BST2 (**Figs 5B**, **5D,** and **S6**). In the case of S$_{omicron}$, the interaction was compromised with the ΔEC1 mutant and slightly impacted with the ΔCC mutant, although it did not reach statistical significance. Similar to S$_{wuhan}$, mutations in EC2, deletion of CT or mutations in the TM domain had no impact on the association with S$_{omicron}$ (**Figs 5C**, **5D,** and **S6**). These observations indicate that the coiled-coil and EC1 domains represent the major surfaces for the S$_{wuhan}$-BST2 interaction, while the EC1 region is

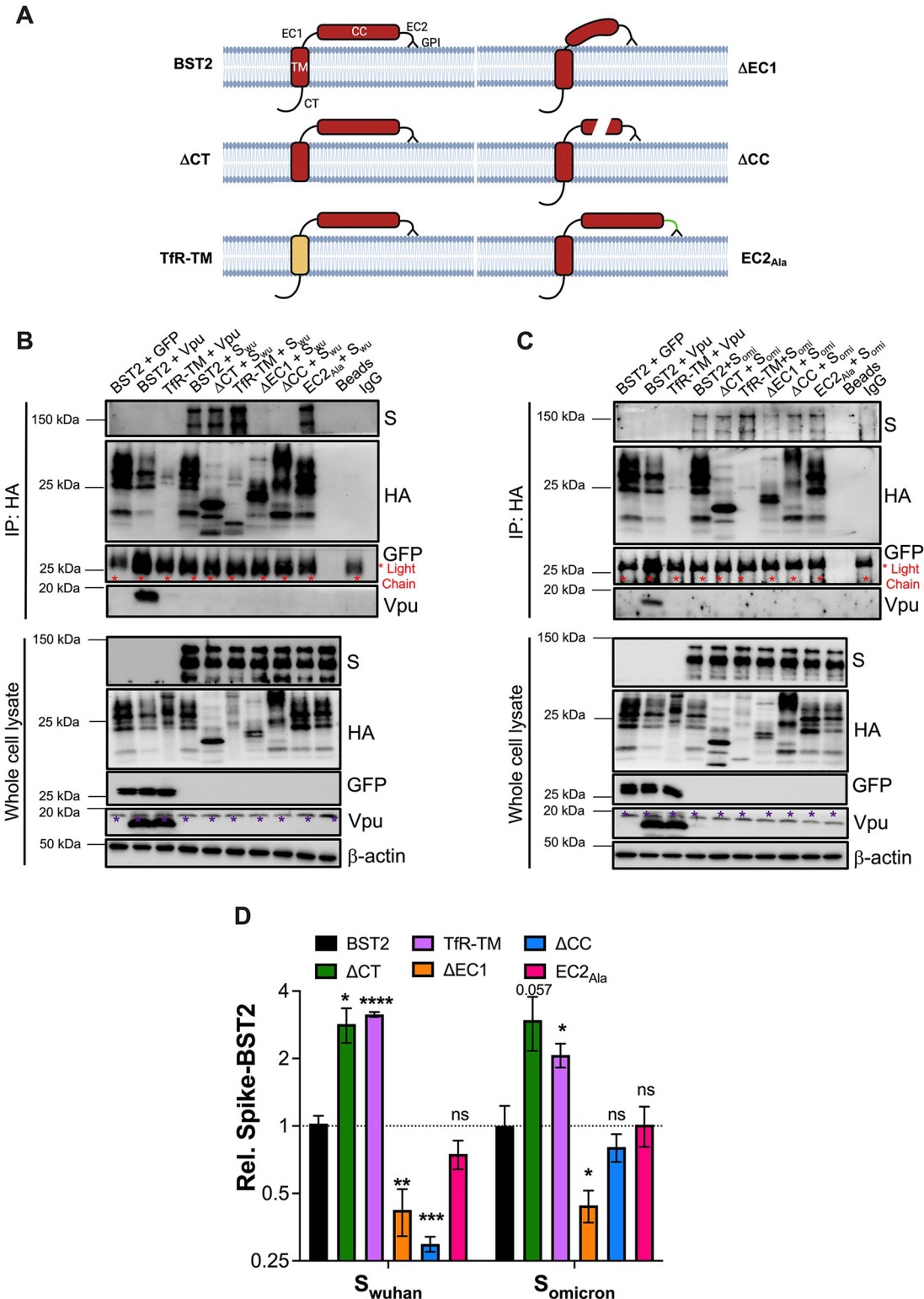

**Fig 5. Multiple surfaces of BST2 are required for Spike binding.** (**A**) Diagram of the architecture of BST2 and BST2 mutants. CT: cytoplasmic tail. TM: transmembrane domain. EC1: extracellular domain region 1. CC: coiled-coil domain. EC2: extracellular domain region 2. GPI: glycosylphosphatidylinositol anchor. The interaction between the Wuhan (**B**) and Omicron (**C**) Spikes and different BST2 mutants was investigated by co-IP. GFP was used as a negative control. HIV-1 Vpu was used as a positive control of a virus protein interacting with BST2. Additional controls included beads only (cell lysates of S and full-length

BST2) and an IgG control (lysis buffer treated with beads coated with anti-HA antibody). ΔCT: BST2 lacking the cytoplasmic tail. TfR-TM: BST2 harboring the transmembrane domain of the transferrin receptor. ΔEC1: BST2 with deletions in the region between the TM and CC domains. ΔCC: BST2 with a truncated coiled-coil domain. EC2$_{Ala}$: BST2 containing Ala substitutions in the region between the CC domain and the GPI anchor. Red asterisks indicate bands corresponding to the light chain of the antibody used in the IP. Purple asterisks correspond to unspecific bands. Blots are representative of 4 independent experiments. (**D**) Relative binding between BST2 and the Wuhan or Omicron Spikes was calculated by densitometry analyses from 4 independent experiments, where binding with full-length BST2 represents 100% (or 1) binding for each Spike variant. *: $p<0.05$, **: $p<0.01$, ***: $p<0.001$, ****: $p<0.0001$, ns: not significant. *BST2 diagram was generated in BioRender.*

more important for the association with S$_{omicron}$. In conclusion, the Spikes use multiple surfaces in the extracellular region of BST2 for their association with this restriction factor.

## The Spike redistributes BST2 to intracellular compartments

BST2 antagonists redirect this protein away from sites of virion assembly through two main mechanisms: either by promoting the degradation of BST2 (i.e., HIV-1 Vpu, KSHV K5, SARS-CoV-1 S and SIV Nef)[17,20,64,65] or by sequestering BST2 in intracellular compartments (i.e., HIV-2 Env, HIV-1 Vpu)[5,66]. To understand how SARS-CoV-2 S antagonizes BST2, its subcellular distribution was examined by fluorescence microscopy. For this, HeLa cells, which endogenously express BST2, were transfected with an empty vector, S$_{wuhan}$ or S$_{omicron}$ and cells were analyzed 48 hours later. Unlike the empty vector-expressing cells, in which most cells displayed a homogenous distribution of BST2 (**Fig 6A**), the overall BST2 expression was diminished in cells expressing either Spike, and its subcellular distribution was altered (**Fig 6B** and **6C**). Notably, while some Spike-expressing cells exhibited redistribution of BST2 to few discrete locations, other cells displayed a more punctuated distribution throughout the cytoplasm. These phenotypes were quantified by calculating the clustering or lacunarity of BST2 from 20 randomly selected cells for each experimental condition (**Fig 6D**). Our results show significantly more clustering of BST2 in Spike-expressing cells. However, no major differences were observed between S$_{wuhan}$ and S$_{omicron}$ (**Fig 6D**).

Next, additional imaging studies were performed using different intracellular markers to reveal the location(s) where BST2 is being targeted. We used ER, *cis*-Golgi, and *trans*-Golgi markers, since along with the plasma membrane, these are the natural locations where BST2 localizes [6,67]. We also utilized CD63, a marker found in late endosomes, exosomes, lysosomes, multivesicular bodies and the plasma membrane [68–71]. Lysosomal markers were also included, since many viruses, including SARS-CoV-1, target BST2 for lysosomal degradation [17,20,64,65]. In cells expressing S$_{wuhan}$, BST2 primarily co-localized with the lysosomal marker, with an average Pearson's $R$ of 0.81. Partial co-localization with ER, *cis*-Golgi and *trans*-Golgi was also observed, averaging a Pearson's $R$ of ~0.4. Remarkably, BST2 seem to distribute opposite to CD63 in cells expressing S$_{wuhan}$ ($R = 0.22$) (**Fig 7A**). Consistent with these findings, BST2 mainly co-localized with lysosomal markers in cells expressing S$_{omicron}$, with an average Pearson's $R$ of 0.72, and partially co-localized with ER and Golgi markers ($R$ ~0.6). CD63 was still the marker that showed lowest co-localization with BST2, with an average Pearson's $R$ of 0.38 (**Fig 7B**). These findings indicate that, similar to SARS-CoV-1 [17], both S$_{wuhan}$ and S$_{omicron}$ redistribute BST2 to lysosomes.

## SARS-CoV-2 Spike promotes the lysosomal degradation of BST2 in a Clathrin- and ubiquitination-dependent manner

To further examine if SARS-CoV-2 causes the lysosomal degradation of BST2, the impact of the Spike on BST2 was investigated in the presence of proteasomal and lysosomal inhibitors. HEK293T-ACE2 cells stably expressing BST2 were transfected with S$_{wuhan}$, S$_{omicron}$ or

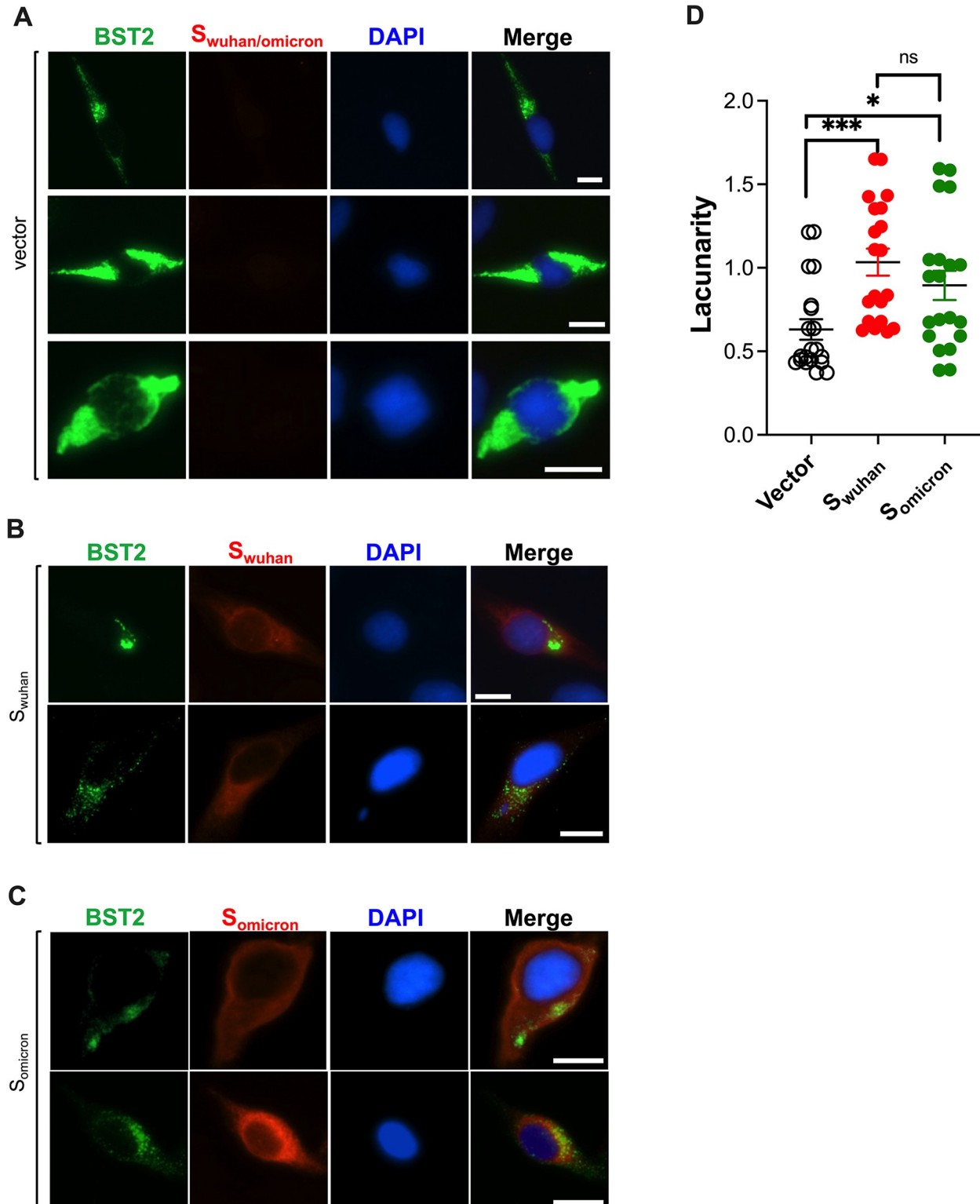

**Fig 6. The SARS-CoV-2 Spike redistributes BST2.** HeLa cells transfected with plasmids encoding an empty vector (**A**), SARS-CoV-2 $S_{wuhan}$ (**B**) or $S_{omicron}$ (**C**) were stained for BST2 (green), Spike (red), and the nuclei (blue). (**D**) The clustering (lacunarity) of BST2 distribution for vector- or Spike-transfected cells was calculated from 20 randomly selected cells per experimental condition. Data correspond to the mean and SEM. 2–3 representative images from 3 independent assays are shown for each experimental condition. Scale bar: 10 μm. *: $p<0.05$, ***: $p<0.001$, ns: not significant.

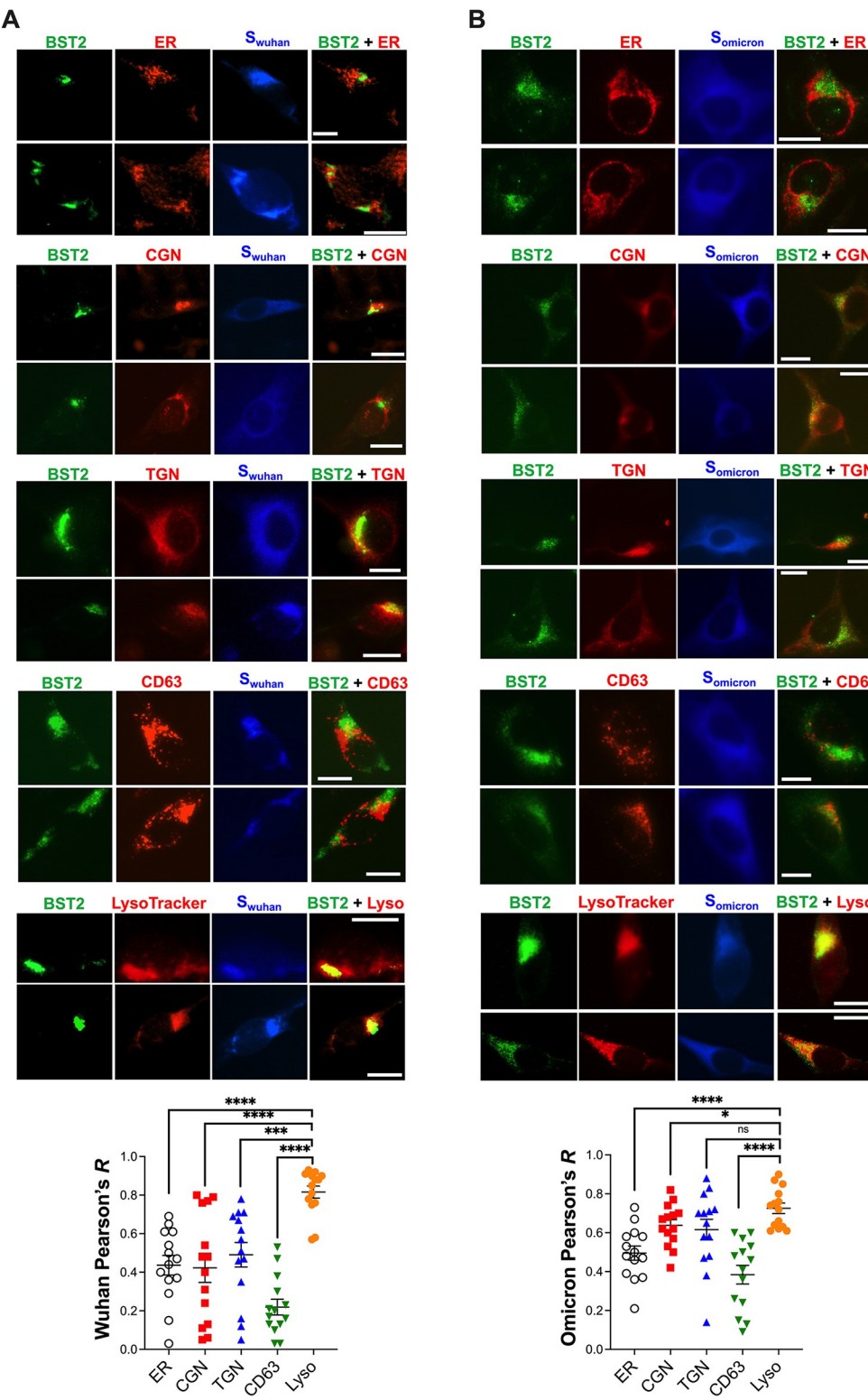

**Fig 7. BST2 distribution overlaps with lysosomal markers in cells expressing SARS-CoV-2 Spike.** The subcellular distribution of BST2 relative to different intracellular markers, including ER, *cis*-Golgi (CGN), *trans*-Golgi (TGN), CD63, and lysotracker, was evaluated by fluorescence microscopy in HeLa cells transfected with the SARS-CoV-2 Wuhan (**A**) and Omicron (**B**) Spikes. The Pearson's correlation coefficient (*R*) for the degree of co-localization between BST2 and each marker was calculated from 14 randomly selected cells. Data correspond to the mean and

SEM. Scale bar: 10 μm. 2 representative images from 3 biological replicates are provided for each experimental condition. *: $p < 0.05$, ***: $p < 0.001$, ****: $p < 0.0001$, ns: not significant.

GST-HA. HIV-1 Vpu was used as a control. Four hours post-transfection, cells were treated with DMSO, MG132 (proteasomal inhibitor) or hydroxychloroquine (Chlor; lysosomal inhibitor) for 44 hours. Cells were then harvested and analyzed for BST2 expression by western blot. A minor enhancement in BST2 levels was observed in cells transfected with GST and treated with these drugs, reflecting that BST2 is naturally degraded through the proteasome and lysosome (**Fig 8A** and **8B**). As previously reported, Vpu caused both proteasomal and lysosomal degradation of BST2 [64], since BST2 levels were rescued in the presence of both inhibitors (**Fig 8A** and **8B**). In line with findings with SARS-CoV-1 [17], BST2 levels were restored in $S_{wuhan}$- and $S_{omicron}$-expressing cells treated with hydroxychloroquine, although a partial rescue, yet insignificant, was observed with MG132 (**Fig 8A** and **8B**). These findings indicate the Spike promotes the lysosomal degradation of BST2.

To understand how the Spike routes BST2 to lysosomes, we explored the role of ubiquitination, since proteins that are targeted to the lysosome are often ubiquitinated [72,73] and many BST2 antagonists recruit E3 ligases to redistribute BST2 [14, 22, 62, 63, 74–76]. This was investigated using TAK-243, a ubiquitination inhibitor [77]. HEK293T-ACE2 cells stably expressing BST2 were transfected with constructs coding for the Wuhan or Omicron Spikes. GST was included as a negative control and HIV-1 Vpu was used as a BST2 antagonist that promotes the ubiquitination of BST2 to cause its degradation [22,63,75,78]. Four hours post-transfection, TAK-243 was added to the cell media and cells were harvested 44 hours later. Addition of TAK-243 rescued BST2 levels in cells expressing the Spike, indicating that ubiquitination is part of the mechanism by which the Spike targets BST2 to the lysosome (**Fig 9A**). Besides ubiquitination, we also explored the role of the endocytosis and autophagy pathways, which are the main mechanisms to target membrane proteins to lysosomes. The role of autophagy was investigated by depleting *ATG5* –an essential gene for autophagosome biogenesis [79,80]–by CRISPR/Cas9 in HEK293T-ACE2 cells stably expressing BST2. Depletion of ATG5 causes an accumulation of the non-lipidated form of LC3 (LC3-I) (**Fig 9B**; blots), which is consistent with a defect in autophagosome elongation. However, no substantial differences in the degree of BST2 downregulation were observed between non-targeting and *ATG5*KD cells expressing the Wuhan or Omicron Spikes, indicating that functional autophagy is dispensable to route BST2 to the lysosome (**Fig 9B**). Next, the role of vesicular trafficking in the downregulation of BST2 was studied using a dominant-negative mutant of the Clathrin adaptor AP180 (AP180C) [63,81] that was co-transfected along with GST, Vpu, $S_{wuhan}$ or $S_{omicron}$ into HEK293T-ACE2 cells stably expressing BST2. Cells transfected with pCGCG, which harbors GFP, were used as controls. In line with previous studies [61,78], Vpu's ability to downregulate BST2 was compromised when Clathrin was not functional. Similarly, the Wuhan Spike could not promote the degradation of BST2. Although BST2 levels were also enhanced in cells co-expressing the Omicron Spike and AP180C, statistically significant differences were still observed compared to the GST-expressing cells (**Fig 9C**), suggesting that, besides Clathrin, the Omicron Spike uses additional mechanisms to downregulate BST2. Overall, these findings indicate that the SARS-CoV-2 Spike uses ubiquitination and Clathrin-dependent trafficking to promote the lysosomal degradation of BST2.

## Discussion

BST2 is a well-studied antiviral restriction factor against many enveloped viruses that traps nascent virions to the cell surface and, therefore, prevents virion release. As for other antiviral

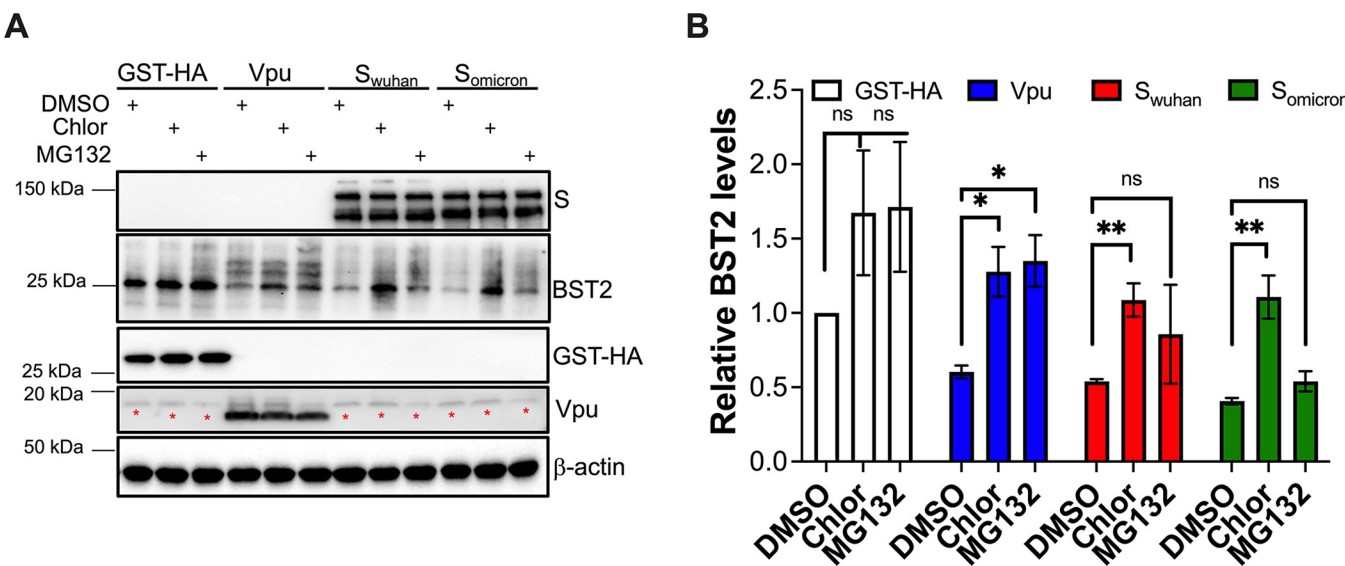

**Fig 8. The Spike causes the lysosomal degradation of BST2.** (**A**) HEK293T-ACE2 cells stably expressing BST2 were transfected with plasmids encoding GST, HIV-1 Vpu, SARS-CoV-2 $S_{wuhan}$, or SARS-CoV-2 $S_{omicron}$. DMSO, MG132, or hydroxychloroquine (Chlor) were added 4 hours post-transfection. BST2 levels were measured by western blot 44 hours later. (**B**) The relative expression of BST2 was calculated by densitometry analyses from 3 biological replicates. Red asterisks indicate unspecific bands. *: $p<0.05$, **: $p<0.01$, ns: not significant. Blots are representative of 3 independent experiments. Data correspond to the mean and SEM of 3 independent experiments.

factors, viruses have evolved mechanisms to counteract this restriction [3,4,12,14,15,17–19,42,44,82,83]. In the specific case of coronaviruses, BST2 interferes with HCoV229E, SARS-CoV-1 and PEDV [15,17,18]. BST2 is also predicted to block SARS-CoV-2 [23,26,27], with different studies proposing several countermeasures by this virus. However, no study to date has directly addressed the impact of BST2 on SARS-CoV-2 during infection and whether the action-counteraction axis between the virus and this restriction factor has changed as new SARS-CoV-2 variants emerge. As for other interferon-stimulated genes (ISGs) [84], *BST2* is induced in response to SARS-CoV-2 infection, indicating that this molecule is part of the antiviral arsenal induced to fight SARS-CoV-2. In line with its effects on other enveloped viruses [4,15,82], BST2 causes a significant reduction in SARS-CoV-2 virion release. However, the virus circumvents this block by downregulating BST2 protein levels. Notably, newly emerged SARS-CoV-2 strains, particularly Omicron, more efficiently downregulate BST2, which increases in turn the production of infectious particles, and may have contributed in part to their high spread. These findings suggest that BST2 antagonism is important for SARS-CoV-2 infection.

The SARS-CoV-2-mediated downregulation of BST2 was observed with endogenous levels of the protein, under conditions of BST2 overexpression (transient transfection and transduction) and by enhancing its expression through interferon (IFN) stimulation. In all these scenarios, Omicron outperformed Hong Kong (HK) at counteracting BST2. However, few differences were noted. First, in A549 cells stably expressing BST2, HK can downregulate BST2 with just MOI = 0.1, and at the same MOI–and with comparable virus protein levels–Omicron downregulates BST2 even further. However, under conditions of IFN-induced BST2, HK only affords its downregulation at MOI = 5. In the case of Omicron, this isolate still performs better than HK, causing noticeable downregulation of BST2 at MOI = 1. While the overall interpretation is the same: Omicron is more efficient than HK at counteracting BST2, we hypothesize that these differences may be caused by the interferon-induced antiviral state,

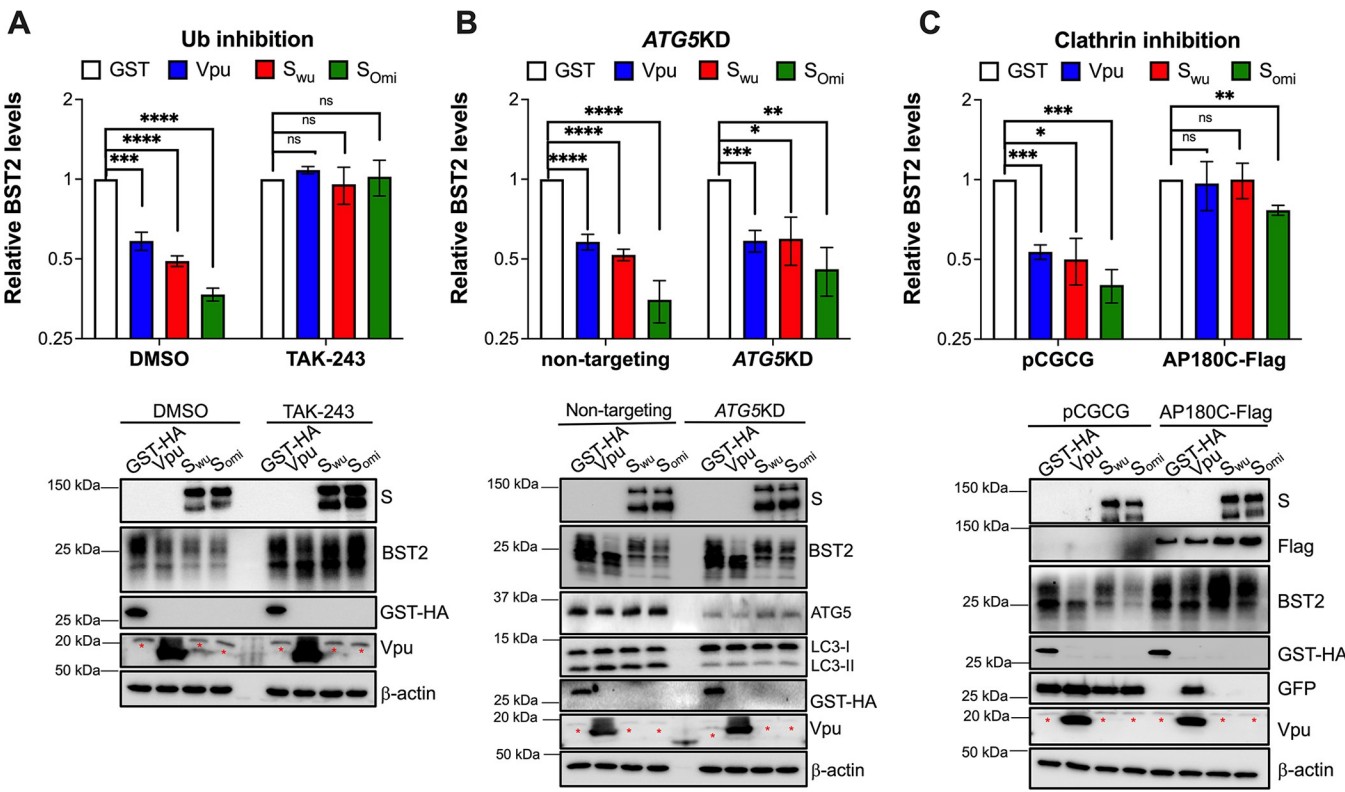

**Fig 9. The Spike uses ubiquitination to route BST2 to lysosomes in a Clathrin-dependent manner.** (**A**) HEK293T-ACE2 cells stably expressing BST2 were transfected with constructs coding for GST, HIV-1 Vpu, $S_{wuhan}$ or $S_{omicron}$. Four hours later, cells were treated with DMSO or the ubiquitination inhibitor TAK-243. Cells were analyzed for BST2 levels 48 hours post-transfection. (**B**) HEK293T-ACE2-non targeting and -ATG5KD cells stably expressing BST2 were transfected with plasmids coding for GST, HIV-1 Vpu, $S_{wuhan}$ or $S_{omicron}$. The levels of BST2 were examined by western blot 48 hours later. (**C**) HEK293T-ACE2 cells stably expressing BST2 were co-transfected with plasmids encoding AP180C-Flag or GFP (pCGCG) along with GST, HIV-1 Vpu, $S_{wuhan}$ or $S_{omicron}$. BST2 levels were measured by western blot 48 hours later. The relative expression of BST2 was calculated by densitometry analyses from 3 biological replicates. Red asterisks indicate unspecific bands. *: $p<0.05$, **: $p<0.01$, ***: $p<0.001$, **** $p<0.0001$, ns: not significant. Blots are representative of 3 independent experiments. Data correspond to the mean and SEM of 3 independent experiments.

which imposes additional obstacles for virus replication, consequently affecting the variants' ability to counteract BST2. Second, the antiviral state triggered by IFNα impacts these two variants differently. While the expression levels of the Spike and Nucleocapsid of HK and Omicron are comparable one day post-infection in the absence of IFNα, the expression of these two proteins was elevated in infections with Omicron in the presence of IFNα, suggesting that, even though this variant causes a higher induction of type I IFN [51], it is more resistant to its effects. In fact, this notion is consistent with previous studies showing that newly emerged isolates, especially Omicron, are more resistant to interferon restriction [85,86].

To identify the BST2 countermeasure, we tested individual SARS-CoV-2 proteins against BST2 and found that S, and to a lesser extent NSP1, downregulate BST2. Similar to SARS-CoV-1 ORF7a, and recent findings with SARS-CoV-2 ORF7a [16,27], we also found that SARS-CoV-2 ORF7a causes the accumulation of a lower molecular band of BST2, which is consistent with a role for ORF7a in preventing BST2 glycosylation [16]. However, despite this effect on BST2 post-translational modifications, SARS-CoV-2 ORF7a caused no net reduction in BST2 expression, indicating that this virus protein is not responsible for the downregulation of BST2. Since ORF7a did not affect overall BST2 protein levels, we focused on NSP1 and Spike. NSP1 is a shutoff factor that blocks host translation and accelerates host mRNA degradation to favor virus gene expression [56,87], so it was not surprising observing a reduction in

BST2 levels. To determine whether NSP1 or S is the main viral gene product that causes the downregulation of BST2, we investigated if these genes have served as hot spots for the accumulation of mutations during the evolution of the virus in humans, or if their expression was enhanced. The reasoning is that our assays with the variants of concern show an improvement in BST2 antagonism and a parallel increase in infectious particle production as the virus adapted in the human host. We confirmed that the expression levels of virus proteins, including NSP1 and Spike, were comparable across the variants of concern tested. So, we next examined for changes in their amino acid sequences. While NSP1 has remained highly conserved, the *Spike* is well known for being the predominant gene mutated in variants of concern. These mutations have been reported to increase binding with the ACE2 host receptor and facilitate immune evasion by becoming resistant to neutralizing antibodies [39,40,88]. Therefore, it seems plausible that the mutations in S also enhance the virus ability to antagonize BST2. To test this hypothesis, we cloned the Spike of the most notorious variants of concern and tested them against BST2. In agreement with our hypothesis, S from these emerged stains, particularly Omicron, downregulate BST2 better than the initial circulating strains, which demonstrates that S is the main BST2 antagonist, and suggests that the virus selects for mutations in S that afford improved anti-BST2 action.

To decipher how S downregulates BST2, we first investigated if these two proteins interact. Consistent with work on SARS-CoV-1 [17], an interaction between the Spike and BST2 was detected. Notably, compared to $S_{wuhan}$, $S_{omicron}$ bound more efficiently to BST2, suggesting that as part of its adaptation to BST2, SARS-CoV-2 is selecting for mutations that improve this physical interaction, either through increased affinity for BST2, augmented avidity for BST2, or by increasing the association with an intermediate factor. Future experiments will address this question. Mutagenesis of the different domains of BST2 revealed that the interaction between the Spike and BST2 requires multiple surfaces in the extracellular region of BST2. Specifically, the coiled-coil (CC) and extracellular region 1 (EC1) play a major role in the association with $S_{wuhan}$, while EC1 is more important for the interaction with $S_{omicron}$. Binding between the Spikes and the EC1 and CC domains of BST2 would require using the S1 region and/or extracellular portion of the S2 region in the Spikes, which are domains that have accumulated most of the mutations identified in the Omicron Spike. While these mutations provide a fitness advantage as explained above, they may have compromised the ability of the Spike to interact with these domains of BST2. However, because $S_{omicron}$ immunoprecipitates more efficiently than $S_{wuhan}$ with BST2 –and $S_{omicron}$ still binds better to the ΔCC-BST2 mutant–we hypothesize that additional mutations in the Omicron Spike have allowed for a more efficient interaction with other regions of BST2. Since deletion of the EC1 domain had the greatest impact on BST2-$S_{omicron}$ binding, we theorize that the Omicron Spike has adapted to favor contact sites surrounding the coiled-coil domain, like EC1. Future experiments will map the residues in the Spikes (Wuhan and Omicron) that facilitate this association with BST2.

The enhanced Spike-BST2 association observed in Omicron likely explains why Omicron promotes a more efficient downregulation of BST2. To understand how the Spike depletes BST2, we studied if it causes differences in the subcellular distribution of this restriction factor. Besides noticing a significant reduction in BST2 levels, we found that Spike-expressing cells displayed a more clustered distribution of BST2. However, two main phenotypes were observed: redistribution of BST2 to few discrete locations in the cytoplasm and more punctuate redistribution. We hypothesize that these differences in BST2 localization represent intermediate steps in the process of downregulating BST2. Additional imaging studies using intracellular markers revealed that BST2 highly overlaps with lysosomal markers, which is in line with the mechanism by which SARS-CoV-1 Spike counteracted BST2 [17]. This notion

was verified using lysosomal and proteasomal inhibitors showing that Wuhan and Omicron Spikes cannot downregulate BST2 in the presence of drugs that disable lysosomal function. Finally, to understand how the Spikes target BST2 to the lysosome, we examined the role of ubiquitination, the autophagy, and the endo-lysosomal pathways. Similar to other BST2 antagonists, the Spike-mediated downregulation of BST2 requires ubiquitination, which suggests that the Spike promotes BST2 ubiquitination, and that this post-translational modification routes the protein for lysosomal sorting. However, we cannot exclude that the Spike or cellular factors are the ubiquitination targets that are required to direct BST2 to the lysosome, since TAK-243 disables ubiquitination in a general manner. In fact, ubiquitination of the Spike has recently been reported [89]. Future studies will investigate the specific role of ubiquitination in this process as well as identify the ubiquitination target(s) and the E3 ligase used by the Spike. Finally, while functional autophagy is dispensable for the downregulation of BST2 by SARS-CoV-2, our assays with AP180C revealed that $S_{wuhan}$ and $S_{omicron}$ use Clathrin-mediated vesicular trafficking to route BST2 to lysosomes. However, a partial decrease in BST2 levels was still observed in cell expressing $S_{omicron}$ under AP180C expression, which likely reflects that this Spike uses additional mechanisms to downregulate BST2. Future studies will investigate this possibility and uncover where (plasma membrane versus intracellular membranes) the Spike attacks BST2. In summary, this study reveals that SARS-CoV-2 uses the Spike to counteract restriction by BST2 and that mutations in the Spike facilitate escape from BST2, suggesting that BST2 antagonism is a contributing factor to the adaptation of SARS-CoV-2 in humans.

## Materials and methods

### Cell lines

Human HEK293T (American Type Culture Collection [ATCC]), HeLa (NIH HIV Reagent Program, Division of AIDS, NIAID, NIH from Dr. Richard Axel), HEK293T-ACE2 (BEI resources), and VeroE6 (ATCC) cells were cultured in complete medium (Dulbecco's Modified Eagle Medium [DMEM, ThermoFisher Scientific] supplemented with 10% fetal bovine serum [FBS, ThermoFisher Scientific], 1% Penicillin-Streptomycin [ThermoFisher Scientific] and 1% L-glutamine [ThermoFisher Scientific]). A549-ACE2 (BEI resources) cells were maintained in complete medium with 50 ng/mL blasticidin S HCl (ThermoFisher Scientific). HEK293T-ACE2 stably expressing pQCXIP-BST2 or pQCXIP, HEK293T-ACE2-BST2-*ATG5*KO and HEK293T-ACE2-BST2-LentiCRISPRv2 cells were grown in complete medium with 1 μg/mL of puromycin (ThermoFisher Scientific). A549-ACE2 cells stably expressing pQCXIP-BST2 or pQCXIP were maintained in complete medium with 50 ng/mL blasticidin S HCl and 1 μg/mL of puromycin. Viability of cells was measured after each transfection. Only cells with viabilities > 90% were considered for further experiments.

### Virus stock production

SARS-CoV-2 propagation was performed at the URMC biosafety level 3 (BSL3) laboratory following the approved standard operating procedures.

- Virus isolates: SARS-CoV-2 Hong Kong, Alpha, Beta, Delta, and Omicron variants were obtained from BEI resources (see Table 1).

- Propagation. 2 x $10^6$ VeroE6 cells were seeded in 10 $cm^2$ dishes. Twenty-four hours later, cells were infected with one of the virus isolates listed above (BEI) at MOI 0.01. Three days later, the supernatants were collected and centrifuged for 10 min at 931 x $g$ to remove cell debris. Next, supernatants were aliquoted in 1 mL cryotubes and stored at -80°C.

**Table 1. Resources.**

| REAGENT | SOURCE | INDENTIFIER |
|---|---|---|
| **Antibodies and dyes** | | |
| **DAPI** | ThermoFisher Sci. | Cat# 62248 |
| **Donkey anti-Mouse (HRP-conjugated)** | Abcam | Cat# ab6885 |
| **Donkey anti-Rabbit (HRP-conjugated)** | Abcam | Cat# ab16284 |
| **LysoTracker Red DND-99** | ThermoFisher Sci. | Cat# L7528 |
| **Mouse AlexaFluor-488 IgG2a** | ThermoFisher Sci. | Cat# A21131 |
| **Mouse AlexaFluor-546 IgG1** | ThermoFisher Sci. | Cat# A21123 |
| **Mouse mAb anti-BST2** | Sigma-Aldrich | Cat# SAB1402131 |
| **Mouse mAb anti-Calnexin** | ThermoFisher Sci. | Cat# MA3-207 |
| **Mouse mAb anti-COSR1** | ThermoFisher Sci. | Cat# MA1-91008 |
| **Mouse mAb anti-GFP** | ThermoFisher Sci. | Cat# MA5-15256 |
| **Mouse mAb anti-HA tag** | BioLegend | Cat# 901502 |
| **Mouse mAb anti-SARS-CoV-2 S (S1-NTD)** | Cell Signaling | Cat# 42172S |
| **Mouse mAb anti-TGN46** | ThermoFisher Sci. | Cat# MA3-063 |
| **Mouse mAb anti-ubiquitin** | ThermoFisher Sci. | Cat# 13–1600 |
| **Mouse mAb anti-β-actin** | Cell Signaling | Cat# 3700S |
| **Rabbit AlexaFluor-350 IgG** | ThermoFisher Sci. | Cat# A11046 |
| **Rabbit mAb anti-BST2** | Cell Signaling | Cat# 19277S |
| **Rabbit mAb anti-BST2** | Abcam | Cat# ab134061 |
| **Rabbit mAb anti-SARS-CoV-2 N** | ThermoFisher Sci. | Cat# MA5-36086 |
| **Rabbit mAb anti-ubiquitin** | Abcam | Cat# ab134953 |
| **Rabbit pAb anti-6X His tag** | Abcam | Cat# ab9108 |
| **Rabbit pAb anti-ATG5** | Cell Signaling | Cat# 2630S |
| **Rabbit pAb anti-NL4-3 Vpu** | HIV Reagent Program | Cat# ARP969 |
| **Rabbit pAb anti-SARS-CoV-2 S** | ThermoFisher Sci. | Cat# PA5-112048 |
| **Rabbit pAb anti-Strep-tag II** | Abcam | Cat# ab76949 |
| **Virus strains** | | |
| **SARS-CoV-2 isolate Hong Kong** | BEI resources | Cat# NR-52282 |
| **SARS-CoV-2 Alpha Variant (B.1.1.7)** | BEI resources | Cat# NR-55461 |
| **SARS-CoV-2 Beta Variant (B.1.351)** | BEI resources | Cat# NR-55282 |
| **SARS-CoV-2 Delta Variant (B.1.617.2)** | BEI resources | Cat# NR-55611 |
| **SARS-CoV-2 Omicron Variant (B.1.1.529)** | BEI resources | Cat# NR-56461 |
| **Chemicals, peptides, and recombinant proteins** | | |
| **2x SDS sample buffer** | Sigma-Aldrich | Cat# S3401 |
| **Acetone** | Sigma-Aldrich | Cat# 270725 |
| **Anti-quenching mounting medium** | Vector Laboratories | Cat# 3304770 |
| **Blasticidin S HCl** | ThermoFisher Sci. | Cat# A1113903 |
| **Chloroform** | Spectrum | Cat# C1220 |
| **Coupling buffer** | ThermoFisher Sci. | Cat# 88805 |
| **DMEM** | ThermoFisher Sci. | Cat# 11885–084 |
| **DMSO** | Sigma-Aldrich | Cat# D2650 |
| **DPBS** | ThermoFisher Sci. | Cat# 14190–144 |
| **Ethanol** | URMC Chemical supply | |
| **FBS** | ThermoFisher Sci. | Cat# 26140–079 |
| **Fish skin gelatin** | Sigma-Aldrich | Cat# 67765 |
| **GenJet *in vitro* DNA transfection reagent** | SignaGen Laboratories | Cat# SL100488 |
| **Goat serum** | ThermoFisher Sci. | Cat# 500062Z |

*(Continued)*

**Table 1.** (Continued)

| REAGENT | SOURCE | INDENTIFIER |
|---|---|---|
| Hydroxychloroquine | Sigma-Aldrich | Cat# H0915 |
| IFNα2a | Sigma-Aldrich | Cat# IF007 |
| iScript cDNA synthesis kit | Bio-Rad | Cat# 1725038 |
| Isopropyl alcohol | Sigma-Aldrich | Cat# W292907 |
| L-glutamine | ThermoFisher Sci. | Cat# 25030–081 |
| Lysis IP buffer | ThermoFisher Sci. | Cat# 87787 |
| Lysis buffer | Sigma-Aldrich | Cat# 4719956001 |
| Methanol | Sigma-Aldrich | Cat# 34860 |
| MG132 | Sigma-Aldrich | Cat# 474791 |
| PBS-tween | Sigma-Aldrich | Cat# P3563 |
| Penicillin-Streptomycin | ThermoFisher Sci. | Cat# 15070–063 |
| Protein G magnetic beads | NEB | Cat# S1430S |
| Puromycin | ThermoFisher Sci. | Cat# A1113803 |
| RayBio COVID-19/SARS-CoV-2 Nucleocapsid Protein ELISA kit | RayBio | Cat# ELV-COVID19N |
| RNase free water | ThermoFisher Sci. | Cat# 10977–025 |
| SsoAdvanced Universal SYBR Green Supermix | Bio-Rad | Cat# 1725272 |
| SuperSignal West Femto maximum sensitivity substrate | Pierce | Cat# 34095 |
| TAK-243 | ThermoFisher Sci. | Cat# 50-187-1707 |
| Triton X-100 | Sigma-Aldrich | Cat# X100 |
| TRIzol Reagent | ThermoFisher Sci. | Cat# 15596018 |
| Trypsin-EDTA (0.25%) | ThermoFisher Sci. | Cat# 25200–056 |
| Experimental models: Cell lines | | |
| HEK293T-ACE2 | BEI resources | Cat# NR-52511 |
| HEK293T-ACE2-BST2 | This study | N/A |
| HEK293T-ACE2-BST2-ATG5KO | This study | N/A |
| HEK293T-ACE2-pQCXIP | This study | N/A |
| A549-ACE2 | BEI resources | Cat# NR-53821 |
| A549-ACE2-BST2 | This study | N/A |
| A549-ACE2-pQCXIP | This study | N/A |
| HEK293T | ATCC | Cat# CRL-11268 |
| HeLa | HIV Reagent Program | Cat# ARP-153 |
| Vero E6 | ATCC | Cat# CRL-1586 |
| Oligonucleotides | | |
| RQ1/RQ2 | Bio-Rad | Cat# qHsaCtlD0001002 |
| gDNA | Bio-Rad | Cat# qHsaCtlD0001004 |
| GAPDH | Bio-Rad | Cat# qHsaCED0038674 |
| Human BST2 | Bio-Rad | Cat# qHsaCID0013844 |
| Recombinant DNA | | |
| CoV2-Spike-D614G-Alpha | Addgene | Cat# 177961 |
| CoV2-Spike-D614G-Beta | Addgene | Cat# 177962 |
| CoV2-Spike-D614G-Gamma | Addgene | Cat# 177963 |
| LentiCRISPRv2-ATG5 | Addgene | Cat# 99573 |
| LentiCRISPRv2 | Addgene | Cat# 52961 |
| MLV Gag-Pol | Addgene | Cat# 14887 |
| pCAGGS-SARS-CoV-2 BA.1 Spike | Addgene | Cat# 185452 |
| pcDNA5-HA-BST2 mutants | This study | N/A |
| pcDNA5-HA-GST | This study | N/A |

(*Continued*)

**Table 1.** (Continued)

| REAGENT | SOURCE | INDENTIFIER |
|---|---|---|
| **pcDNA5-His-SARS-CoV1-S** | This study | N/A |
| **pcDNA5-His-SARS-CoV2-S Wuhan** | This study | N/A |
| **pCGCG** | Dr. Jacek Skowronski | N/A |
| **pCGCG-HIV-1-Vpu** | Dr. David T. Evans | N/A |
| **pCMV-AP180C-Flag** | Dr. David T. Evans | N/A |
| **pGFP-C-Lenti** | Origene | Cat# TR30021 |
| **pLKO.1-sh-scramble** | ThermoFisher Sci | Cat# TRCN00000010871 |
| **pQCXIP** | Clontech | Cat# 631516 |
| **pQCXIP-BST2** | This study | N/A |
| **pRetroQ-AcGFP1-N1** | Clontech | Cat# 632506 |
| **psPAX2** | HIV Reagent Program | Cat# ARP-11348 |
| **SARS-CoV2-ORFs** | [91] | N/A |
| **VSV-G** | Addgene | Cat# 12259 |
| **Software and algorithms** | | |
| **ChemiDoc Image Lab** | Bio-Rad | Version 6.1.0 |
| **FlowJo www.flowjo.com Version 10.7.1** | | |
| **GraphPad Prism** | www.graphpad.com | Version 10 |
| **MacVector** | www.macvector.com | Version 18.1.3 |
| **Gen5** | BioTek Instruments | Version 3.14 |

Mock preparation. $5 \times 10^6$ HEK293T cells were seeded in 10 cm$^2$ dishes. Twenty-four hours later, cells were transfected with 3.75 μg psPAX2 packing plasmid, 1.25 μg VSV-G-expressing plasmid, and 5 μg of a lentiviral vector encoding GFP (pGFP-C-Lenti; Origene). Forty-eight hours later, the supernatant was collected and centrifuged for 10 min at 931 $x\,g$ to remove cell debris. Next, the culture supernatant was aliquoted in 1 mL cryotubes and VLPs were cryopreserved at -80°C. These VLPs were used as mock in infections.

SARS-CoV-2 virus-like particles (VLPs) preparation. $6 \times 10^6$ HEK293T cells were seeded in 10 cm$^2$ dishes. Twenty-four hours later, cells were co-transfected with 5 μg of pLVX-EF1alpha-SARS-CoV-2-M, 5 μg of pLVX-EF1alpha-SARS-CoV-2-N, 2.5 μg of pLVX-EF1alpha-SARS-CoV-2-E and 2.5 μg of pLVX-EF1alpha-SARS-CoV-2-S. Forty-eight hours later, the supernatant was collected and centrifuged for 10 min at 931 $x\,g$ to remove cell debris. Next, the culture supernatant was aliquoted in 1 mL cryotubes and VLPs were cryopreserved at -80°C. An aliquot of VLPs was pelleted and examined by western blot together with pelleted preparations of SARS-CoV-2 HK and Omicron of known titers, so their approximate concentration could be inferred. These VLPs were used as a control for abortive infection at MOI ~ 1.

## Virus infection assay

All SARS-CoV-2 infection experiments were performed at the URMC BSL3 laboratory following the approved standard operating procedures.

Infection of BST2 stable cell lines. $10^6$ HEK293T-ACE2 and A549-ACE2 cells stably expressing pQCXIP or pQCXIP-BST2 were seeded in 6-well plates. Twenty-four hours later, cells were infected with different SARS-CoV-2 strains at MOI = 0.1 or 1 (see Table 1 for virus strains). As controls, we included untreated cells (NT) and mock-infected cells, which consisted of lentiviral-like particles harboring GFP. Twenty-four hours post-infection, supernatants were

collected to assess virion production and infectivity, and cells were harvested by adding lysis buffer (Sigma-Aldrich) supplemented with 0.1% Triton-x-100 (Sigma-Aldrich).

Infection of cell lines treated with interferon. $5 \times 10^5$ A549-ACE2 cells were seeded in 12-well plates. Twenty-four hours later, cells were infected with SARS-CoV-2 HK or Omicron at MOI 0.1, 1 or 5. As controls, we included untreated cells (NT) and cells infected with SARS-CoV-2 VLPs at MOI ~ 1. One-hour later, cells were washed, supplemented with fresh media, and treated with either DMSO or IFNα2a (1,000 U/mL; Sigma-Aldrich). Twenty-four hours post-infection, cells were harvested by adding lysis buffer (Sigma-Aldrich) supplemented with 0.1% Triton-x-100 (Sigma-Aldrich) and analyzed by western blot.

## Median Tissue Culture Infectious Dose Assay (TCID$_{50}$)

$2.5 \times 10^4$ cells (VeroE6, HEK293T-ACE2 or A549-ACE2) were seeded in 96-well plates. Stock viruses as well as culture supernatants recovered from SARS-CoV-2 infections were serially diluted ($10^{-1}$ to $10^{-9}$) in DMEM with 3% FBS. Cell media were removed, and cells were subsequently infected with 100 µL of virus dilutions in six replicates. Three days post-infection, the cytopathic effect (CPE) on each well was determined by optical microcopy. The TCID$_{50}$ of viruses was calculated using the Spearman-Kärber method [90].

## ELISA

Culture supernatants from SARS-CoV-2 infections were also examined by ELISA to determine the concentration of SARS-CoV-2 virions produced from pQCXIP- and pQCXIP-BST2-expressing cells (either HEK293T or A549 cells). For this, RayBio COVID-19/SARS-CoV-2 Nucleocapsid Protein ELISA kit was used following the manufacturer's instructions (RayBio). Virion production was expressed as the percentage of virus release, where cells expressing pQCXIP were considered to afford 100% of viral release.

## Plasmid constructs

Transfection of plasmids was achieved using GenJet *in vitro* DNA transfection reagent (Signa-Gen Laboratories, Ijamsvile, MD) following the manufacturer's instructions.

- Plasmids encoding BST2: Human BST2 (full-length: 180 amino acids) was cloned into the expression vector pcDNA3 [19]. BST2 was subcloned into pcDNA5 and the retroviral vector pQCXIP. BST2 truncated mutants, including ΔCT (a mutant without the cytoplasmic tail, lacking the first 20 amino acids of the protein), ΔCT+TM (a mutant without both the cytoplasmic tail and transmembrane domains, which only harbors amino acids 46–180), BST2-TfR-TM (a mutant in which the BST2 transmembrane domain was replaced by the transmembrane domain of the human Transferrin Receptor), ΔEC1 (a mutant with a 50-amino acid deletion between residues 51–101 in the extracellular domain), ΔCC (a mutant with a 54-amino acid deletion between residues 101–154 in the extracellular coiled-coil domain), and EC2$_{Ala}$ (a mutant with six alanine substitutions between residues 155 and 160, located at the end of the coiled-coil domain and before the signal for the GPI anchor) were cloned into pcDNA5. All constructs have an HA tag in their N-terminus.

- Plasmids encoding SARS-CoV-2 ORFs: codon-optimized constructs encoding each open reading frame of SARS-CoV-2 were obtained from a lentiviral library (pLVX-EF1alpha) generated by the Krogan laboratory [91] and deposited to Addgene. Each gene is tagged with 2xStrep tags in the C-terminus, and the plasmids harbor an IRES Puromycin selection marker. Codon-optimized SARS-CoV-1 Spike (S) and SARS-CoV-2 Wuhan-1 S were cloned into pcDNA5 with a 6xHis-tag in their C-terminus. Expression vectors harboring the *S* gene

for other circulating SARS-CoV-2 variants were purchased from Addgene (Alpha, Beta and Gamma were deposited by Dr. Melanie Ott [92]; Delta was deposited by Dr. Daniel Conway; and Omicron was deposited by Marceline Cote [93]).

- Plasmid encoding HIV-1 Vpu: Codon optimized HIV-1 NL4-3 Vpu was cloned into the expression vector pCGCG, and was a gift from Dr. David T. Evans, University of Wisconsin, Madison, WI. The pCGCG expression vector harboring EGFP was a gift from Dr. Jacek Skowronski, Case Western Reserve University, Cleveland, OH. This plasmid was used as a negative control.

- Plasmid encoding HA-GST: GST was cloned into the expression vector pcDNA5 with an HA-tag in its N-terminus [94].

- Dominant-negative constructs: A plasmid encoding the dominant-negative mutant of AP180 (AP180C-Flag) was a gift from Dr. David T. Evans, University of Wisconsin, Madison, WI.

- CRISPR/Cas9 constructs. In order to knockdown *ATG5*, a CRISPR/Cas9 plasmid targeting this gene, LentiCRISPRv2-ATG5, was obtained through Addgene. This plasmid was deposited by Dr. Edward Campbell [95]. As negative control, cells were transduced with Lenti-CRISPRv2, which was deposited by Dr. Feng Zhang [96].

- Plasmids for the generation of retrovirus-like particles (VLPs). To generate retroviral-like particles, a plasmid encoding the Moloney Murine Leukemia Virus (MLV) Gag-Pol polyprotein was used. The plasmid was deposited to Addgene by Dr. Tannishtha Reya [97]. To generate lentiviral-like particles, the lentiviral packaging construct psPAX2 (NIH HIV Reagent Program) was used. The expression vector coding for the envelope glycoprotein of VSV, pMD2-G, was deposited to Addgene by Dr. Didier Trono.

## Generation of cells stably expressing BST2 and *ATG5*KD cells

- Generation of virus-like particles (VLPs). $5 \times 10^6$ HEK293T cells were seeded in 10 cm$^2$ dishes. Twenty-four hours later, cells were transfected with 3.75 μg packing plasmid (either MLV Gag-Pol for retroviral-like particles or psPAX2 for lentiviral-like particles), 1.25 μg VSV-G-expressing plasmid, and 5 μg of retroviral or lentiviral vectors encoding our gene of interest, including pQCXIP, pQCXIP-BST2, LentiCRISPRv2 or LentiCRISPRv2-ATG5. Forty-eight hours later, the culture supernatant was collected and centrifuged for 10 min at 931 x *g* to remove cell debris. Next, the cleared supernatant was aliquoted in 1 mL cryotubes and VLPs were cryopreserved at -80˚C.

- Transduction. $3 \times 10^6$ HEK293T-ACE2 and A549-ACE2 cells were seeded in 25 cm$^2$ flasks. Twenty-four hours later, cells were transduced with 2 mL of pQCXIP, pQCXIP-BST2, or 1 mL of pQCXIP-BST2 plus either 1 mL of LentiCRISPRv2 or LentiCRISPRv2-ATG5 VLPs at 37˚C for 2 hours. Forty-eight hours later, cells were transduced with a second round of VLPs. One day after this second transduction, the cell medium was replaced and supplemented with 1 μg/mL of puromycin. The stable cell lines were verified after 2 weeks of puromycin selection by western blot and flow cytometry.

## Western blot

Cells subjected for western blotting analysis were washed with DPBS (ThermoFisher Scientific) and harvested by adding lysis IP buffer (ThermoFisher Scientific). For

SARS-CoV-2-infected cells, lysis buffer was supplemented with 1% Triton X-100. Cells were then kept on ice for 30 min. Cell debris was removed by centrifugation at 16,000 x $g$ at 4˚C for 8 min. Next, the supernatants were mixed with 2x SDS sample buffer (Sigma-Aldrich), and samples were boiled for 10 min on a heat block. Proteins were then separated using 12% SDS-PAGE polyacrylamide gels. Proteins were then transferred to a polyvinylidene difluoride (PVDF) membrane (Bio-Rad) using a Trans-Blot Turbo Transfer System (Bio-Rad, Hercules, CA). Membranes were incubated for 1 hour with blocking buffer (Bio-Rad) at room temperature, followed by an overnight incubation with primary antibodies (see Table 1) at 4˚C. Next, membranes were washed 3 times with PBS-tween (Sigma-Aldrich) followed by 1 hour incubation with the secondary antibodies (Table 1) at room temperature. Subsequently, three additional washes in PBS-tween were performed before imaging the membranes. Finally, membranes were developed by adding SuperSignal West Femto maximum sensitivity substrate (Pierce), and proteins were visualized in a ChemiDoc imaging system (Bio-Rad). The expression level of proteins was quantified using ChemiDoc Image Lab software (Bio-Rad) and normalized to that of β-actin. Each experiment was repeated three independent times.

## Co-immunoprecipitation assay

$10^6$ HEK293T cells were seeded on 6-well plates. Twenty-four hours later, cells were co-transfected with 2 μg of either pCGCG-Vpu, pCGCG, pcDNA5, pCDNA5-GST-HA, pcDNA5-SARS-CoV-1-S, pcDNA5-SARS-CoV-2-S, pCAGGS-SARS-CoV-2 BA.1 Spike, or pLVXEF1alpha-SARS-CoV-2-ORF7a and 1 μg of either pcDNA3, pcDNA3-BST2, pcDNA5 or pcDNA5-BST2-HA constructs. Forty-eight hours post-transfection, cells were washed with DPBS and harvested using lysis IP buffer. Cells were then kept on ice for 30 min. Cell debris was removed by centrifugation at 16,000 x $g$ at 4˚C for 8 min. Next, the whole cell lysates were pre-cleared by incubating them with protein G magnetic beads (New England Biolabs) for 1 hour at 4˚C, which removed any unspecific binding. In parallel, fresh protein G beads were coated with the antibody of interest (anti-BST2 mouse IgG$_{2a}$, 1:100; anti-HA, 1:100; Table 1) for 1 hour at room temperature, followed by three washes with coupling buffer (ThermoFisher Scientific) to remove excess antibody. Next, the pre-cleared lysates were incubated with the antibody-coated protein G beads overnight at 4˚C. Next, beads were washed with lysis IP buffer 5 times. Finally, the beads were resuspended in 2x SDS sample buffer and the samples were analyzed by western blotting. As controls, a sample with beads only and a sample consisting of IP lysis buffer mixed with beads and antibody (IgG control) were included. These controls helped rule out any unspecific bands detected by western blot that corresponded to the IgG heavy or light chains or material from the magnetic beads. Each experiment was repeated three independent times.

## Protein degradation assay

$9 \times 10^5$ HEK293T-ACE2 cells stably expressing pQCXIP-BST2 were seeded in 6-well plates. Twenty-four hours later, cells were transfected with 3 μg of plasmid encoding for GST, HIV-1 Vpu, SARS-CoV-2 Wuhan S or Omicron S. Four hours post-transfection, proteasomal (MG132, 0.3 μM; Sigma-Aldrich) or lysosomal (hydroxychloroquine, 60 μM; Sigma-Aldrich) inhibitors were added. DMSO (Sigma-Aldrich) was included as a control. Cells were harvested 44 hours later by adding lysis IP buffer. Protein expression was analyzed by western blot and densitometry analyses were performed to quantify protein levels. Each experiment was repeated three independent times.

## Ubiquitination inhibition assay

9 x $10^5$ HEK293T-ACE2 cells stably expressing pQCXIP-BST2 were seeded in 6-well plates. Twenty-four hours later, cells were transfected with 3 µg of plasmid encoding GST, HIV-1 Vpu, SARS-CoV-2 Wuhan S or Omicron S. Four hours post-transfection, the ubiquitination inhibitor TAK-243 (200 nM; ThermoFisher Scientific) was added. DMSO was included as a control. Cells were harvested 48 hours post-transfection by adding lysis IP buffer. Protein expression was analyzed by western blot and densitometry analyses were performed to quantify protein levels. Each experiment was repeated three independent times.

## Reverse transcription followed by quantitative PCR (RT-qPCR)

- RNA extraction and cDNA synthesis.

$10^6$ A549-ACE2 cells were seeded in 6-well plates. Twenty-four hours later, cells were infected with SARS-CoV-2 HK at MOI = 1 (see Table 1 for virus strains). Mock infection and untreated cells were included as controls. Six hours later, cells were washed with DPBS, and total RNA was extracted by adding 1 mL Trizol (ThermoFisher Scientific) per well. 200 µL of chloroform (Spectrum) was then added, and samples were centrifuged at 12,000 x $g$ for 15 min at 4°C to create three phases of separation: lower red phenol-chloroform phase, an interphase, and a transparent upper aqueous phase. The aqueous phase was collected and mixed with 500 µL of isopropyl alcohol to precipitate RNA. RNA was then washed with 75% ethanol and eluted in RNase free water (ThermoFisher Scientific). The RNA concentration and A260/A280 ratios were measured in a NanoDrop (ThermoFisher Scientific). Next, 1 µg of purified RNA was converted into cDNA using the iScript cDNA synthesis kit (Bio-Rad), following the manufacturer's instructions.

- qPCR.

To measure the mRNA levels of *BST2*, the SYBR green-based real-time qPCR method was employed. For each sample, different controls including RNA quality (RQ1 and RQ2), genomic DNA contamination (gDNA), and housekeeping gene (GAPDH) were measured by qPCR. In each PCR reaction, 5 µL 2x SsoAdvanced Universal SYBR Green Supermix (Bio-Rad), 0.1 µL cDNA, 4.4 µL RNase free water (ThermoFisher Scientific), and 0.5 µL primer pairs for BST2/control gene were included. The amplification program was as follows: 2 min at 95°C for initial activation, 40 cycles at 95°C for 5 seconds, 60°C for 30 seconds, and then melting analyses from 65 to 95°C (0.5°C increments). Each sample was analyzed by qPCR in two technical replicates. Experiments were performed three independent times for each experimental condition. All primers are listed in Table 1.

## Flow cytometry

HEK293T-ACE2 cells stably expressing pQCXIP or pQCXIP-BST2 were infected with either mock VLPs or SARS-CoV-2 HK at MOI = 1. Twenty-four hours later, cells were trypsinized and collected in flow tubes. Cells were washed 3 times by centrifugation with DPBS at 500 x $g$ for 5 min and were permeabilized by using FIX & PERM cell permeabilization kit (ThermoFisher Scientific), following the manufacturer's instructions. Next, cells were stained with an anti-BST2 mouse $IgG_{2a}$ primary antibody (Table 1) at a 1:60 ratio for 20 min at room temperature. After staining, cells were washed 3 times by centrifugation with DPBS at 500 x $g$ for 5 min. Next, cells were incubated with a secondary antibody anti-mouse $IgG_{2a}$ Alexa Fluor 488 (Table 1) at a 1:500 ratio for 20 min at room temperature. After the secondary incubation, cells were washed 3 additional times and fixed with 6% paraformaldehyde (Sigma-Aldrich). Fixed

cells were stored at 4°C for 24 hours at the BSL3 facility before imaging. Data was collected on a BD Accuri C6 Plus Flow Cytometer (BD Biosciences, Franklin Lakes, NJ) and analyzed by FlowJo. Each experiment was repeated three independent times.

## Immunofluorescence microscopy

$9 \times 10^5$ HeLa cells were seeded in 6-well plates. Twenty-four hours later, cells were transfected with 3 μg of pcDNA5, SARS-CoV-2 Wuhan S, or SARS-CoV-2 Omicron S. Six hours later, cells were trypsinized and re-seeded at a concentration of $5 \times 10^4$ per well in sterile tissue culture-treated 8-well chamber slides (ThermoFisher Scientific). Forty-eight hours post-transfection, cells were washed with ice cold DPBS three times. Next, cells were fixed and permeabilized by adding 50:50 acetone:methanol (Sigma-Aldrich) for 10 min at -20°C. Cells were then blocked with antibody diluent solution (2% fish skin gelatin [Sigma-Aldrich] + 0.1% Triton X-100 [Sigma-Aldrich] + 10% goat serum [ThermoFisher Scientific] with 1 x DPBS) for 30 min at room temperature followed by an incubation of 1 hour with a primary antibody cocktail at room temperature (anti-BST2 mouse IgG$_{2a}$, anti-His-tag rabbit IgG, and anti-cellular markers mouse IgG1 –ER, Golgi, and CD63 –at a dilution of 1:60, 1:200, 1:200, respectively; see Table 1). Next, cells were washed with wash buffer (2% fish skin gelatin + 0.1% Triton X-100 with 1 x DPBS) for three times and incubated with a secondary antibody cocktail (Alexa-488 anti-mouse IgG$_{2a}$, Alexa-350 anti-rabbit IgG, and Alexa-546 anti-mouse IgG$_1$ at 1:500; see Table 1) for 30 min. For some immunofluorescence studies, cells were also incubated with DAPI (ThermoFisher Scientific, 1:5000; see Table 1) for 5 min to visualize the nuclei. After this step, the slides were washed and mounted using anti-quenching mounting medium (Vector Laboratories). The slides were visualized using a BioTek Lionheart FX automated microscope using 20x and 40x objectives and filter cubes 377, 469, and 586 nm. Images were processed and analyzed using the Gen5 software (BioTek Instruments, Winooski, VT). Person's correlation coefficients ($R$) and lacunarity were calculated using Fiji [98].

## Quantification and statistical analysis

Statistical calculations were performed using two-tailed unpaired Student's T test analyses. All statistical analyses were performed using Graph Pad Prism version 10.0.0. p values $\leq 0.05$ were considered statistically significant.

## Supporting information

**S1 Fig. The degree of BST2 expression in A549-ACE2-BST2 cells is comparable to that afforded by IFNα stimulation.** The levels of BST2 in A549-ACE2 cells engineered to constitutively express BST2 were compared to HeLa cells, which express BST2 endogenously, and parental A549-ACE2 cells treated with IFNα. For this, A549-ACE2 cells were treated with 100, 1,000 and 10,000 U/mL of IFNα2. Cells were harvested 24 hours later and analyzed by western blot.
(TIFF)

**S2 Fig. The SARS-CoV-2 variants of concert counteract BST2 more efficiently than SARS-CoV-2 HK.** (**A**) HEK293T-ACE2 and (**B**) A549-ACE2 cells stably expressing pQCXIP or pQCXIP-BST2 were infected with SARS-CoV-2 HK, Alpha, Beta, Delta, or Omicron variants at MOI = 0.1 or 1. Twenty-four hours post-infection, the levels of BST2 and virus proteins were measured by western blot, and infectious virion production was measured by TCID$_{50}$ (bottom panels). *: $p<0.05$, **: $p<0.01$, ns: not significant. Blots are representative of 3

biological replicates. Data correspond to the mean and SEM of 3 independent experiments.
(TIFF)

**S3 Fig. Evolution of the SARS-CoV-2 Spike.** Schematic representation of the SARS-CoV-2
Spike. Mutations accumulated in variants of concern compared to the Wuhan sequence
(NC_045512) are indicated. Spike sequences were obtained from NCBI GenBank: Alpha
B.1.1.7: MZ344997.1, Beta B.1.351: MW598419.1, Gamma P.1: MW642250.1, Delta B.1.617.2:
MZ009823.1, Omicron B.1.1.529: OL672836.1.
(TIFF)

**S4 Fig. ORF7a and NSP1 are not hot spots for mutations in variants of concern.** Sequence
alignments of ORF7a (**A**) and NSP1 (**B**) across variants of concern. SARS-CoV-2 ORF7a and
NSP1 sequences were obtained from NCBI GenBank; Wuhan-Hu-1: NC_045512.2, Alpha
B.1.1.7: MZ344997.1, Beta B.1.351: MW598419.1, Delta B.1.617.2: MZ009823.1, Omicron
B.1.1.529: OL672836.1
(TIFF)

**S5 Fig. Mutations accumulated in the Spike of variants of concern afford more efficient
antagonism of BST2.** (**A**) HEK293T-ACE2 cells stably expressing BST2 were transfected with
plasmids encoding the *Spike* gene from each of the indicated variants of concern. BST2 and
Spike levels were measured by western blot. (**B**) Relative BST2 expression was calculated by
densitometry analyses, normalized to actin, and expressed as the percentage of BST2. Red
asterisks indicate unspecific bands. **: $p < 0.01$, ***: $p < 0.001$, ****: $p < 0.0001$. Blots are repre-
sentative of 3 biological replicates. Data correspond to the mean and SEM of 3 independent
experiments.
(TIFF)

**S6 Fig. Multiple surfaces of BST2 are required for Spike-mediated antagonism.** Extended
data for Fig 5. (**A**) Diagram of the architecture of BST2 and BST2 mutants. CT: cytoplasmic
tail. TM: transmembrane domain. EC1: extracellular domain region 1. CC: coiled-coil domain.
EC2: extracellular domain region 2. GPI: glycosylphosphatidylinositol anchor. (**B**-**E**) The
interaction between the Wuhan and Omicron Spikes and different BST2 mutants was investi-
gated by co-IP. GFP was used as a negative control. HIV-1 Vpu was used as a positive control
of a membrane virus protein interacting with BST2. Additional controls included beads only
(cell lysates of S$_{omicron}$ and full-length BST2) and an IgG control (lysis buffer treated with
beads coated with anti-HA antibody). ΔCT: BST2 lacking the cytoplasmic tail. ΔCT+TM:
BST2 lacking the cytoplasmic and transmembrane domains. TfR-TM: BST2 harboring the
transmembrane domain of the transferrin receptor. ΔCC: BST2 with a truncated coiled-coil
domain. ΔEC1: BST2 with deletions in the region between the TM and CC domains. EC2$_{Ala}$:
BST2 containing Ala substitutions in the region between the CC domain and the GPI anchor.
Red asterisks indicate bands corresponding to the light chain of the antibody used in the IP.
Yellow asterisks indicate bands corresponding to the antibody. Purple asterisks correspond to
unspecific bands. Blue pound symbol indicates bands that correspond to the ΔCC mutant.
Blots are representative of 3 independent experiments. *BST2 diagram was generated in BioRen-
der.*
(TIFF)

**S1 Information. All unedited blots shown in this manuscript have been compiled and
annotated in one single PDF.**
(PDF)

**S2 Information. All the numerical data used to build the graphs in this manuscript are provided in an annotated excel file.**
(XLSX)

## Acknowledgments

We want to thank Drs. Martin Pavelka and Sonia Rosenberger as well as the BSL3 core facility for providing training and allowing us to use the BSL3 laboratory.

## Author Contributions

**Conceptualization:** Yuhang Shi, Sydney Simpson, Ruth Serra-Moreno.

**Formal analysis:** Yuhang Shi, Sydney Simpson, Ruth Serra-Moreno.

**Funding acquisition:** Ruth Serra-Moreno.

**Investigation:** Yuhang Shi, Yuexuan Chen, Haley Aull, Jared Benjamin.

**Methodology:** Yuexuan Chen, Haley Aull, Jared Benjamin.

**Project administration:** Ruth Serra-Moreno.

**Supervision:** Ruth Serra-Moreno.

**Writing – original draft:** Yuhang Shi.

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
