## [Decision Letter · Decision Letter 0]

30 Oct 2023

Dear Dr. Serra-Moreno,

Thank you very much for submitting your manuscript "Mutations accumulated in the Spike of SARS-CoV-2 variants of concern allow for more efficient counteraction of the restriction factor BST2/Tetherin" for consideration at PLOS Pathogens. As with all papers reviewed by the journal, your manuscript was reviewed by members of the editorial board and by several independent reviewers. In light of the reviews (below this email), we would like to invite the resubmission of a significantly-revised version that takes into account the reviewers' comments.

This study explores how SARS-CoV-2 may counteract a crucial host restriction factor involved in suppressing other viruses and is of timely interest. 

Your manuscript was reviewed by 4 individuals who all generally praised the depth of the work and see value in this study’s findings. However, the reviewers raised several concerns that will need to be addressed moving forward. Reviewers 1 and 2 noted several, albeit minor, inconsistencies between the conclusions and the molecular data. Reviewer 3 and 4 have suggestions for additional experiments assessing endogenous tetherin levels and interferon stimulation. Several reviewers requested quantification of the image data for more robust analysis. All reviewers requested additional clarification with the domain mapping experiments and data presentation.

We cannot make any decision about publication until we have seen the revised manuscript and your response to the reviewers' comments. Your revised manuscript is also likely to be sent to reviewers for further evaluation.

Sincerely,

Michael Letko, PhD

Academic Editor

PLOS Pathogens

Sonja Best

Section Editor

PLOS Pathogens

Kasturi Haldar

Editor-in-Chief

PLOS Pathogens

orcid.org/0000-0001-5065-158X

Michael Malim

Editor-in-Chief

PLOS Pathogens

orcid.org/0000-0002-7699-2064

Thank you for submitting your manuscript exploring tetherin counteraction by SARS-CoV-2 spike. This study explores how SARS-CoV-2 may counteract a crucial host restriction factor involved in suppressing other viruses and is of timely interest.

Your manuscript was reviewed by 4 individuals who all generally praised the depth of the work and see value in this study’s findings. However, the reviewers raised several concerns that will need to be addressed moving forward.

Reviewers 1 and 2 noted several, albeit minor, inconsistencies between the conclusions and the molecular data. Reviewer 3 and 4 have suggestions for additional experiments assessing endogenous tetherin levels and interferon stimulation. Several reviewers requested quantification of the image data for more robust analysis. All reviewers requested additional clarification with the domain mapping experiments and data presentation.

Reviewer's Responses to Questions

**Part I - Summary**

Reviewer #1: The manuscript describes that SARS-CoV-2 S proteins can downregulate BST-2 through clathrin-dependent endocytosis by binding its multiple surfaces in the extracellular domain, which results in ubiquitination-dependent lysosomal degradation, and that the S-mediated BST-2 downregulation is more efficient in recent variants than in early isolates, suggesting the continuous adaptation of SARS-CoV-2 in human host cells. The experiments are well designed, and the results are of interest. Still, there are a few concerns that need to be addressed, as follows.

Reviewer #2: In this manuscript, Shi et al. investigate the mechanism of BST2/tetherin counteraction by SARS-CoV-2 Spike and explore how the ongoing evolution of Spike influences this function. BST2 is an interferon stimulated gene (ISG) that restricts enveloped viruses by tethering them to cells upon budding. Several reports have demonstrated restriction of SARS-CoV-2 by BST2, but viral counteraction has focused on the role of ORF7a. SARS-CoV-1 Spike is known to counteract BST2, but little has been reported on the role of SARS-CoV-2 Spike. The authors show that SARS-CoV-2 infection increases BST2 expression in A549s, while higher MOI infection in cell lines overexpressing BST2 leads to BST2 degradation, an effect which may be stronger in more recent variants of concern. The authors demonstrate that this degradation occurs through lysosomal re-localization in a Clathrin-dependent manner, and suggest this is driven by multiple physical interactions on the extracellular surface.

The work is important and fills a gap in our understanding of BST2 counteraction by SARS-CoV-2. The effect of Spike on BST2 levels is striking (though most assays rely on overexpression) and the lysosomal degradation data are clear. Some of the physical interaction/domain mapping work needs clarification and a more robust presentation of the imaging data is required. While there does appear to be a difference in BST2 degradation between early and late variants, this could be due to many factors. The ‘enhanced affinity‘ argument is not convincing with the data as presented and the evolutionary argument cannot really be made with the data on hand. Overall, major revisions are needed to address methodological and interpretation issues before this manuscript can be recommended for publication.

Reviewer #3: The studies by Shi and co-authors examined the interactions between BST2 and SARS-CoV-2, and demonstrated that BST2 reduces SARS-CoV-2 virion release, and the virus uses S protein to downregulate BST2. They also demonstrated a direct interaction between S and BST2, which routes BST2 for lysosomal degradation in a Clathtin- and Ubiquitin-dependent manner. The studies are extensive and largely well-done. There are a few minor issues:

1) the statement of SARS-CoV-2 variants, and that each variant is more efficient than the previously circulating strain at counteracting BST2. The data to support this statement is weak, and not very strong, also there are only very limited variants examined.

2) Fig. 2. The authors have shown that the omicron has a striking, much strongest phenotype in dimmish BST2. Omicrons are more infectious and likely replicate faster and infect more cells. Would the strong degradation of BSL2 be caused by higher levels of replication rather than a stronger capacity of the variant virus per se ? If we look at data on Fig. 3B, when S protein was used alone, Swuhan and Somicron have comparable levels of BST2.Please clarify the differences between the virus and the S protein in antagonizing BST2

3) the statement “Whereas BST2 was initially recognized as an antiviral factor for its ability to tether HIV

virions to the host cell membrane…”

This is Not an accurate statement. BST2 was first identified to be a membrane protein downregulated by KSHV and HIV Vpu, suggesting that it is likely an antiviral factor (Bartee, E., McCormack, A., and Fruh, K. 2006, Quantitative membrane proteomics reveals new cellular targets of viral immune modulators. PLoS Pathog. 2, e107. 10.1371/journal.ppat.0020107). The work led to studies for its anti-HIV activity later (please see, Damme, et al. 2008, Cell Host & Microbe 3, 245-252). Please acknowledge and cite the earlier PLoS Pathogen paper.

Reviewer #4: In this manuscript, Shi et al. endeavor to elucidate a potential mechanism by which SARS-CoV-2 counteracts host antiviral factor bone marrow stomal antigen (BST2). BST2 is a type I interferon stimulated gene (ISG) that restricts the release of budding enveloped virus particles including HIV-1, other retroviruses, arenaviruses, influenza viruses, herpesviruses, filoviruses, and several coronaviruses.

To elucidate whether BST2 also restricts SARS-CoV-2 virus and whether SARS-CoV-2, in turn, counteracts BST2, they performed various experiments using HEK-293T-ACE2+ and A549-ACE2+ cells stably expressing BST2 and different variants of SARS-CoV-2. The authors conclude that 1) BST2 is induced in response to SARS-CoV-2 infection and, in turn, SARS-coV-2 downregulates BST2 protein levels, 2) SARS-CoV-2 variants of concern have improved their anti-BST2 activity, 3) SARS-CoV-2 uses the Spike (S) protein to downregulate BST2, 4) multiple binding sites of BST2 are required for S-mediated downregulation of BST2, and 5) SARS-CoV-2 S promotes lysosomal degradation of BST2 in a Clathrin- and Ubiquitin-dependent manner. This conclusion is based on five key pieces of data:

1) Infection of A549-ACE2+ cells stably expressing BST2 with SARS-CoV-2 Hong Kong (HK) variant and measuring the BST2 expression 6 hours post infection (hpi) by RT-qPCR. Infection of HEK293T-ACE2+ cells stably expressing BST2 with SARS-CoV-2 HK variant and analysing protein levels of BST2 24 hpi by Western blot.

2) Infection of HEK293T-ACE2+ and A549-ACE2+ cells stably expressing BST2 with SARS-CoV-2 HK, Alpha, Beta, Delta, or Omicron variants and analysing protein levels of BST2 24 hpi by Western blot.

3) Transient transfection of HEK293T-ACE2+ cells stably expressing BST2 with constructs expressing individual SARS-CoV-2 proteins. BST2 protein levels were analysed by Western blot.

4) Generation of a series of BST2 truncation and point mutants which were used to co-transfect HEK293T cells with a plasmid expressing either S-Wuhan or S-Omicron. Co-IP was used to determine their ability to interact with S-Wuhan and S-Omicron.

5) Transfection of HEK293T-ACE2+ cells stably expressing BST2 with plasmid expressing either S-Wuhan or S-Omicron. 4 hours post transfection (hpt), cells were treated with either MG132 (proteasomal inhibitor), hydroxychloroquine (lysosomal inhibitor), or TAK-243 (ubiquitination inhibitor) for 44 hours and BST2 protein levels were analysed by Western blot. The role of vesicular trafficking in the downregulation of BST2 was studies using a dominant-negative mutant of Clarthrin (AP180C) that was co-transfected with either S-Wuhan or S-Omicron into HEK293T-ACE2+ cells stably expressing BST2. BST2 protein levels were analysed by Western blot.

**Part II – Major Issues: Key Experiments Required for Acceptance**

Reviewer #1: Major points

1) Lines 167-168: Although the authors’ state that ORF7a caused an enrichment of a lower molecular band of BST2, Fig. 3A does not show this phenotype (probably due to the overexposure?).

2) Lines 289-290: The authors statement “deletion of the EC1 region drastically reduced the association with BST2 and S-wuhan and caused a partial loss in binding with S-omicron” is inconsistent with the results showing that deltaEC1 is still sensitive to degradation mediated by these S proteins (see Whole cell lysate in the same figure).

3) Fig. 7D and 7E: Because CD63 is the marker of the late endosomal and lysosomal membranes, its colocalization with BST-2 (Fig. 7D) should not be too different from that in lysotracker (Fig. 7E).

Reviewer #2: Major considerations:

1. Degradation of BST2 in the context of infection or S overexpression is only demonstrated in cell lines overexpressing BST2. To provide convincing evidence that Spike-mediated BST2 degradation is relevant to infection, revisions should demonstrate infection- and S-mediated degradation of endogenous BST2 in a lung epithelial cell line such as A549 or Calu-3, or ideally in primary lung epithelial cells. The non-BST2 overexpressing lanes in Figure S1 may be sufficient given proper exposure times. Cells may need to undergo treatment with exogenous IFN to induce suitable BST2 levels for blotting, which would need to be paired with higher MOI infection. This reviewer would also recommend inclusion of a remdesivir treated control in this experiment to demonstrate that BST2 downregulation is dependent on infection and de novo Spike expression and is not triggered by the challenge itself.

2. It would also be helpful to compare endogenous BST2 expression in non-overexpressing cell lines before and after IFN treatment to that shown in the overexpressing lines used in the paper, to contextualize how these lines compare to what we might expect to see in vivo.

3. The data regarding enhanced degradation by each successive variant of concern is complicated and the argument that this is due to enhanced affinity (Figure 4B) is not overly convincing. In the Figure 2 infection panels, Spike and infection levels vary despite uniform challenge due to the many dynamic factors influencing replication, Spike expression, and Spike localization/trafficking. These factors could easily impact the BST2 degradation phenotype. Indeed, the Figure 3C experiment that attempts to fix these issues by direct transfection of the different Spike plasmids shows a reduced effect, maybe only a two-fold difference between Wuhan and Omicron Spike proteins, and many of the other variants seem equivalent in contrast to the stepwise evolution argument (though this would require stats to determine). Looking at Spike expression relative to degradation (Figure 3B), Alpha even seems to perform worse on a per molecule basis. Given that no VOC evolved from any previous VOC, the evolution argument doesn’t really make conceptual sense either. This reviewer would recommend focusing the main text on looking at the Wuhan and Omicron Spike proteins only, relegating the other VOCs to the supplement. This should be paired with a thorough revision of the paper to remove any evolutionary argument, though this is something that could be speculated on in the Discussion. As the authors acknowledge, there were many changing selective pressures on SARS-CoV-2, including adaptive immunity, various non-pharmaceutical interventions, and a host of other ISGs and other facets of innate immunity. Discussion of any role BST2 may have had in SARS-CoV-2 Spike adaptation should be rooted in that context.

4. The use of affinity purification to claim a physical interaction between membrane proteins is similarly complicated, but the proper controls are included and acceptable (though this caveat should be mentioned). However, the claim that Omicron improves BST2 degradation through enhanced affinity is reliant on a minor change in band intensity using this approach in Figure 4B. This would be much more convincing if shown by an orthogonal method.

5. The domain mapping experiments in Figure 5 lead to somewhat mixed results. The delta Ct truncation seems to consistently improve binding and/or degradation while the delta EC and delta CC domains do seem more resistant to this degradation. The issue is that the results are presented so piecemeal and over so many different blots with different exposures, it is exceptionally hard to interpret. This section would be massively improved by a repeat of this experiment with only the most relevant constructs included and run on a single gel. The rest could be shunted to supplement.

6. The imaging data in Figures 6 & 7 needs to be quantified more convincingly. The BST2 distribution with the Omicron Spike protein looks distinctively different from that of the Wuhan Spike in Figure 6; the Omicron cells should be quantified alongside the Wuhan cells in Figure 6b. Furthermore, an objective metric is needed to quantify clustering, such as lacunarity (can be measured using FracLac in ImageJ) or even the percentage of pixels above some BST2 signal threshold in each cell. The BST2 signal seems to be quite bright in both the homogenous and clustered distributions, so this type of analysis could further emphasize that the decrease in BST2 levels occurs with the clustered localization. For Figure 7, it is necessary to quantify colocalization between BST2 and the various compartment markers. It would also be beneficial to perform this analysis for the Omicron Spike to further support claims of differential BST2 interactions between the different Spike proteins. Microscopy showing BST2 localization to the lysosome in SARS-CoV-2 infected cells would be ideal, though the requirement for a BSL3 environment may make this challenging.

Reviewer #3: (No Response)

Reviewer #4: • All figures need better resolution in particular all Western blots.

• Fig. 3A: How is Spike different form SARS-1 S or SARS-2 S shown in the right blot of this panel? This is not clear and should be clarified. Also, why is effect of HIV-1 Vpu so subtle in the right blot compared to the left one? Beta-actin seems comparable between the blots.

• Lines 258-260: Include Fig. reference in the text. Whilst very subtle, there seem to be interaction between HIV-1 Vpu and HA-CTdel-tm BST2. There is a clear HIV-1 Vpu signal on the WB; therefore, the statement in line 259 is inaccurate and should be changed accordingly.

• Fig. 6B: Clarify in the plot or its figure legend which Spike was quantified (Wuhan or Omicron?). It would be good to see a quantification of both, Wuhan and Omicron Spike, separately. Add this information.

• Line 309: I disagree. The distribution of BST2 (based on the two represented confocal images presented) seems different in the presence of S-Omicron from that in the presence of S-Wuhan. It is more like that of the BST2 in the presence of the control plasmid. Please include quantification as in Fig. 6B for S-Omicron as well.

• Line: 315-316 and Fig. 7C: This does not seem to be a minor overlap but rather similar to that observed in Fig. 7E. All panels in this figure need quantification to determine co-localisation. Additionally, it would be good to perform the same experiment with S-Omicron to see the differences.

**Part III – Minor Issues: Editorial and Data Presentation Modifications**

Reviewer #1: Minor points

1) Lines 98-99 “BST2 can be found in several compartments including ER, Golgi and the ERGIC”: Citation required.

2) Lines 120-121 and Fig. 1A: Showing the other ISGs’ data is preferable.

3) Line 143: “targeted by virus pathogens” -> “targeted by virus proteins”?

4) Lines 160- : There are no descriptions of NSP8, NSP9, and ORF8.

5) Lines 246-247 “Here, we did not use the GST-HA control since the BST2 constructs were HA-tagged”: An unnecessary sentence.

6) Lines 277-280: Although the authors state “the quantifications for the relative Spike-BST2 binding showed an enhancement in binding with deltaCT (Figure 5F; green bars), likely due to the remarkable low levels of the deltaCT mutant in these samples (Figure 5B; lanes 7 and 8), this explanation is difficult to understand.

7) Fig. 3A legend: It should be described what tags were used for NSP1, NSP8, NSP9, ORF7a, and ORF8.

8) Fig. 5: The panels of S in cell lysates (Fig. 5B) and in IP (Fig. 5D) should be aligned in parallel with those of HA.

9) The authors should discuss the recent paper by Stewart et al (EMBO Report, e57224. 2023).

Reviewer #2: Minor considerations:

1. The levels of progeny virus as measured by TCID50 shown in infection of 293-ACE2 in Figures 1 & 2 are much lower than the input virus, and there is no post-infection rinse step stated in the methods. This raises concerns about cell line permissivity and potential effects being driven by the challenge itself. For this reason, we found Figure S1 more compelling than Figure 2 and would recommend swapping them.

2. N ELISA on supernatant does not strictly measure viral particles, as N can be secreted (López-Muñoz et al. 2022, DOI: 10.1126/sciadv.abp9770). Clarify this in discussion of Figure 1b.

3. For bar graphs, try to include individual data points overlaid rather than error bars alone.

4. For immunofluorescence data, it might be worth reconsidering some of the color choices to make it easier to discern detail. For single channels, greyscale or green can be much easier to pick out detail than blue. For two channel images, green and magenta are preferable to green and red, which are indistinguishable to the colorblind.

5. The authors frequently stray into lengthier discussions of the data or speculation that should be shunted to the Discussion (i.e., lines 168-179).

6. The authors should proof their language to remove qualitative modifiers (i.e., ‘remarkable’ in line 221, ‘dramatic’ line 235, ‘remarkable’ line 279, etc).

Reviewer #3: (No Response)

Reviewer #4: • Fig. 3.B: There is no explanation of what the red star represents in the HIV-1 Vpu blot. I assume it’s non-specific binding. This should be defined.

• Fig. 5B-5E: Label S with Wuhan as you have labelled S-Omicron for a clarification and consistency.

• Fig. 5F: Symbols for a statistical significance are missing on the green (CTdel) and light purple (TfR-TM) bars. Please add.

• Fig. 6A: What are three rows in top panel and what are the two rows in bottom panel? I assume these are just replicates, but it needs to be clarified. Same for Fig. 6B, what are the two rows?

• Fig. 7: Lack of consistency. Label S as in Fig. 6, i.e., S-Wuhan. As in Fig. 6 it is unclear what are the two rows in each panel (probably replicates, but this needs to be clarified).

• Fig. 8A and B, and Fig. 9 (all panels): Lack of consistency. Label S-Wuhan as S-Omicron.

PLOS authors have the option to publish the peer review history of their article (what does this mean?). If published, this will include your full peer review and any attached files.

Reviewer #1: No

Reviewer #2: No

Reviewer #3: **Yes: **Yuntao Wu

Reviewer #4: No
---

## [Editor Report · Decision Letter 1]

19 Dec 2023

Dear Dr. Serra-Moreno,

We are pleased to inform you that your manuscript 'Mutations accumulated in the Spike of SARS-CoV-2 Omicron allow for more efficient counteraction of the restriction factor BST2/Tetherin' has been provisionally accepted for publication in PLOS Pathogens.

Best regards,

Michael Letko, PhD

Section Editor

PLOS Pathogens

Sonja Best

Section Editor

PLOS Pathogens

Kasturi Haldar

Editor-in-Chief

PLOS Pathogens

orcid.org/0000-0001-5065-158X

Michael Malim

Editor-in-Chief

PLOS Pathogens

orcid.org/0000-0002-7699-2064
---

## [Editor Report · Acceptance letter]

3 Jan 2024

Dear Dr. Serra-Moreno,

We are delighted to inform you that your manuscript, "Mutations accumulated in the Spike of SARS-CoV-2 Omicron allow for more efficient counteraction of the restriction factor BST2/Tetherin," has been formally accepted for publication in PLOS Pathogens.

Best regards,

Michael Malim

Editor-in-Chief

PLOS Pathogens

orcid.org/0000-0002-7699-2064